

# Global atmospheric CO₂ inverse models converging on neutral tropical land exchange but diverging on fossil fuel and atmospheric growth rate

Benjamin Gaubert[1], Britton B. Stephens[1], Sourish Basu[2], Frédéric Chevallier[3], Feng Deng[4], Eric A. Kort[5], Prabir K. Patra[6], Wouter Peters[7], Christian Rödenbeck[8], Tazu Saeki[9], David Schimel[10], Ingrid Van der Laan-Luijkx[7], Steven Wofsy[11], and Yi Yin[12]

[1]National Center for Atmospheric Research, Boulder, CO,USA
[2]University of Colorado Boulder and NOAA Earth System Research Laboratory Boulder, CO, USA
[3]Laboratoire des Sciences du Climat et de l'Environnement, Institut Pierre-Simon Laplace, CEA-CNRS-UVSQ, Gif sur Yvette, 91191 CEDEX, France
[4]Department of Physics, University of Toronto, Canada
[5]University of Michigan, Ann Arbor, MI, USA
[6]RGGC/IACE/ACMPT, Japan Agency for Marine-Earth Science and Technology (JAMSTEC), Yokohama 236 0001, Japan
[7]Meteorology and Air Quality, Wageningen University, Wageningen, Netherlands
[8]Max Planck Institute for Biogeochemistry, 07745 Jena, Germany
[9]Center for Global Environmental Research, National Institute for Environmental Studies, Tsukuba, Japan
[10]Jet Propulsion Laboratory, California Institute of Technology, Pasadena, CA, USA
[11]Harvard University, Cambridge, MA, USA
[12]California Institute of Technology, Pasadena, CA, USA

**Correspondence:** Benjamin Gaubert (gaubert@ucar.edu)

**Abstract.** We have compared a suite of recent global CO₂ atmospheric inversion results to independent airborne observations and to each other, to assess their dependence on differences in northern extratropical vertical transport and to identify some of the drivers of model spread. We evaluate posterior CO₂ concentration profiles against observations from the High-Performance Instrumented Airborne Platform for Environmental Research (HIAPER) Pole-to-Pole Observations (HIPPO) aircraft cam-

paigns over the mid Pacific in 2009-2011. Although the models differ in inverse approaches, assimilated observations, prior fluxes, and transport models, their latitudinal distributions of land fluxes have converged significantly since the Atmospheric Carbon Cycle Inversion Intercomparison (TransCom3) and the REgional Carbon Cycle Assessment and Processes (RECCAP) projects, with model spread reduced by 80 % since TransCom3 and 70 % since RECCAP. Most modelled CO₂ fields agree reasonably well with the HIPPO observations, in particular for annual mean vertical gradients in the northern hemisphere.

Northern hemisphere vertical mixing no longer appears to be a dominant driver of northern versus tropical annual flux differences. Our newer suite of models still gives modest northern extratropical land uptake and near neutral tropical land uptake for 2009-2011, thus implying a continued strong uptake in intact tropical forests given estimates of emissions from deforestation. The results from these models for other time periods (2004-2014, 2001-2004, 1992-1996), and re-evaluation of the TransCom3 Level 2 and RECCAP results confirms that tropical land carbon fluxes including deforestation have been near neutral for sev-

eral decades. However, models still have large disagreements on ocean-land partitioning, and this is influenced by differences



in prescribed fossil fuel emissions and is associated with differences in retrieved atmospheric growth rate. The fossil fuel and the atmospheric growth rate terms have been thought to be the best-known terms in the global carbon budget, but we show that they dominate the model spread at the largest scales and currently limit our ability to assess regional scale terrestrial fluxes and ocean-land partitioning from the model ensemble.

## 1  Introduction

Current appraisals of the global atmospheric carbon budget are informed by surface fluxes computed by inverse transport models (e.g., Newsam and Enting, 1988; Tans et al., 1990; Rayner et al., 1999; Gurney et al., 2002, 2003, 2004; Peylin et al., 2013). Net carbon flux to the atmosphere is derived from temporal and spatial $CO_2$ gradients given by atmospheric observations and prior estimates of component fluxes and their uncertainties. This assessment of atmospheric sources and sinks relies on 1) atmospheric tracer transport models that link fluxes to atmospheric $CO_2$ fields, 2) prior emissions and sinks (e.g., from process model flux estimates), 3) the spatial and temporal representativeness and coverage of the observational network, and 4) error statistics associated with each information piece. Since the problem is underdetermined, it is essential to quantify the uncertainty and biases of posterior fluxes and $CO_2$ concentrations with independent observations and cross-model comparisons. The most prominent community-wide inverse result intercomparison that included comparisons of posterior concentrations to independent observations was the TransCom 3 study (Gurney et al., 2002, 2004), which studied fluxes for the 1992-1996 period. This comparison could focus on the impact of transport model differences by optimizing the fluxes using a common method over the same regions (11 land and 11 ocean). One particular feature of the seasonally resolved (Level 2) TransCom 3 inversions (hereafter denoted as T3L2) was the direct dependence of flux estimates on vertical gradients of $CO_2$ (Stephens et al., 2007), leading to a different partitioning between Northern Extratropical (NE) versus Tropical (T) land sinks.

A more recent community-wide $CO_2$ inverse model intercomparison was carried out as part of the REgional Carbon Cycle Assessment and Processes Project (RECCAP, https://www.globalcarbonproject.org/reccap; Canadell et al. 2011). The atmospheric inversion component of RECCAP was a comprehensive intercomparison that analyzed long-term mean, long-term trend, interannual variations and mean seasonal variations of $CO_2$ fluxes using common post-processing (Peylin et al., 2013). In RECCAP, the 11 inverse models used different inversion techniques, atmospheric models, and observational datasets. When the fluxes were analyzed for the years 2001 to 2004, Peylin et al. (2013) found an overall improved consistency between inversions at large scale and over specific regions compared to T3L2 when the network of atmospheric sites was less dense. RECCAP inversions showed a general agreement on the total natural land carbon flux long-term mean and its interannual variability over 1991-2010. The total ocean plus land sink estimates were more robust over the NE than for the Tropics and in the Southern Extratropics (SE). The remaining spread led to a disagreement on the NE/T/SE land partitioning, with some models simulating a stronger tropical source compensated by larger NE and SE sinks. Peylin et al. (2013) also noted that the group of models that assimilated observations at their corresponding times rather than using monthly means had more consistent, weaker tropical sources, and weaker northern sink land fluxes.

Several additional inverse modelling intercomparison studies have more recently involved satellite, surface and joint surface-





satellite inversion (e.g., Chevallier et al., 2014; Houweling et al., 2015). In these studies, the inversion systems used space borne retrievals of column-average dry air-mole fraction of $CO_2$ (XCO2) from the Orbiting Carbon Observatory-2 (OCO-2) satellite since July 2014 (Eldering et al., 2017) and from the Greenhouse Gas Observing SATellite (GOSAT; Kuze et al. 2009) instrument since January 2009. Those inverse exercises, however, are still sensitive to satellite retrieval algorithms and the in-

versions' prior assumptions. In particular, the results are sensitive to systematic errors from transport (Houweling et al., 2010) and satellite retrievals (Chevallier, 2015).

Schimel et al. (2015) investigated the NE versus T + SE land flux partitioning as indicated by atmospheric inversions, biosphere process model simulations, and forest inventory estimates, and estimated a large land uptake over the tropics by intact forests due to a significant $CO_2$ fertilization effect. This study argued for the importance of comparing posterior $CO_2$ fields to observa-

tions, which was not done in RECCAP, in order to fully understand and predict terrestrial land sinks, and their variation due to $CO_2$ and climate feedbacks. A follow-up inversion intercomparison focused on East Asia and found that large flux adjustments were possible even though models well simulated the observed gradient in vertical profiles measured by aircraft, because the uncertainties from model transport and fossil fuel (FF) prior emissions were compensated by the flux adjustments (Thompson et al., 2016).

The HIPPO campaign (Wofsy, 2011), spanned large latitudinal, vertical and temporal coverage (2009 to 2011) and provides a useful atmospheric trace gas data set for investigating the consistency of inverse fluxes and posterior concentration results. Graven et al. (2013) found an increase of the $CO_2$ seasonal amplitude by up to 50 % at mid to high latitudes of the northern hemisphere and at altitudes ranging between 3 and 6 km between the HIPPO period and the 1950s. Deng et al. (2015) compared posterior $CO_2$ and $O_3$ fields from GEOS-Chem to the HIPPO observations to diagnose the impact of the upper troposphere

and lower stratosphere (UTLS) definition on retrieved fluxes. These results indicate a significant impact of transport errors on retrieved fluxes. Frankenberg et al. (2016) evaluated the CarbonTracker CT2013B and Monitoring Atmospheric Composition and Climate MACC v13r1 atmospheric inverse models, and satellite retrievals from GOSAT, TES (Tropospheric Emission Spectrometer) and AIRS (Atmospheric Infrared Sounder) in comparison to HIPPO measurements.

After years of continuous model development, the goal of this study is to investigate whether global inverse models are still

highly dependent on northern hemisphere vertical transport errors and on a priori assumptions.

Our two main approaches to answer this question are:

– First, we compare modelled $CO_2$ after flux optimization to independent aircraft in-situ $CO_2$ observations from the HIPPO campaign (2009-2011).

– Second, we compare the observationally-constrained fluxes for latitudinal bands across models and on a global scale

both across models and to budget estimates provided by the Global Carbon budget 2016 (Le Quéré et al., 2016).

Measurements and inversion systems are described in Sect. 2. In Sect. 3.1, we present the results of the comparison of mod-eled posterior $CO_2$ vertical gradients with HIPPO measurements. In Sect. 3.2, we analyze the differences in the meridional distribution of land sinks and global carbon estimates for the years 2009 to 2011 from inverse modelling of atmospheric in-situ observations together and with GCP2016. In Sect. 3.3, we compare inverse model and GCP2016 estimates at the global scale,



including prescribed fossil and retrieved atmospheric growth rate terms. Conclusions and a summary of the findings are given in Sect. 4.

## 2 Methods

### 2.1 Participating models

The list of participating inverse models is shown in Table 1 and more details are available in the supplement. These include 10 different inverse modeling systems or system variants. The inversion systems differ in many aspects such as transport models, wind fields, analysis procedures, and subset of assimilated observations. The ACTM system performed 2 inversions with different prescribed fossil fuel (FF) emissions (Saeki and Patra, 2017), one based on totals from the Carbon Dioxide Information Analysis Center (CDIAC; Boden et al. 2016 and another based on the International Energy Agency (IEA 2016),

which allows us to assess sensitivity to the FF prior only. This is also the case for the 2 Carbon Tracker Europe versions, CTE2016-FT (Fast Track) and CTE2017-FT, where only the subset of observations and the FF prior are different (van der Laan-Luijkx et al., 2017).

It is worth noting that some inverse models are constructed in a similar framework. Some share the same transport model, such as TM5 that is used in 4 inversions, and some use the same meteorological fields. Five inverse systems nudge their forecast field

to the ERA-Interim reanalysis (Dee et al., 2011). The two longest flux estimates, from CAMS (v16r1) and Jena (s85_v4.1), are used to reproduce the comparison with observations as in (Stephens et al., 2007) over the T3L2 period (1992 to 1996). The Jena s85_v4.1 and s04_v4.1 inversions differ in their calculation periods and station sets used: Jena s85_v4.1 starts in 1985 using only 23 stations that cover this entire period, while s04_v4.1 uses many more sites (59) and starts in 2004. This also allows us to separate the impact of the number of sites assimilated over the most recent period.

### 2.2 The Global Carbon Budget 2016

The Global Carbon Project (GCP) gathers observational and model-based flux estimates from multiple organizations and research groups around the world to yearly report a global budget of atmospheric $CO_2$ (Le Quéré et al., 2016). The GCP Carbon Budget version 2016 (hereafter denoted GCP2016; Le Quéré et al. 2016) is the most recent version with flux estimates forced

to balance globally. The most recent version (GCP2017; Le Quéré et al. 2018) separated an explicit unknown ocean or land flux term, which prevents simple comparisons of the type presented here. Specifically, the land/ocean partitioning in GCP2016 is based on multiple observational constraints on the ocean flux (SOCEAN) for the 1990s, extrapolated forward with a suite of 7 global ocean models. As pointed out in Le Quéré et al. (2018), there are considerable uncertainties in this extrapolation, with the estimated ocean/land partitioning for later decades dependent on the models.

The GCP2016 atmospheric growth rate GATM is derived from atmospheric $CO_2$ measurements at marine boundary layer (MBL) sites made by the US National Oceanic and Atmospheric Administration Earth System Research Laboratory (NOAA/ESRL;



| Acronym | References | Grid Spacing | Fossil Fuel priors | Transport Model | Number of vertical layers | Meteorological fields | Available Period |
|---|---|---|---|---|---|---|---|
| CAMS (v16r1) | Chevallier et al. (2005, 2010)* | 3.75° x 1.875° | CDIAC/GCP2016 | LMDZ | 39 | ERA interim | 1979 to 2016 |
| Jena (s04_v4.1) | Rödenbeck et al. (2003) Rödenbeck (2005) | 4°x5° | CDIAC | TM3 | 19 | NCEP | 2004 to 2016 |
| Jena (s85_v4.1) | // | 4°x5° | CDIAC | TM3 | 19 | NCEP | 2004 to 2016 |
| CTE2016-FT | van der Laan-Luijkx et al. (2017) | 1°x1° | CDIAC | TM5 | 25 | ERA interim | 2001 to 2015 |
| CTE2017-FT | // | 1°x1° | CDIAC | TM5 | 25 | ERA interim | 2000 to 2016 |
| CT2016 | Peters et al. (2007)** | 1°x1° | ODIAC v2016 and "Miller" | TM5 | 25 | NCEP | 2001 to 2015 |
| ACTM-IEA | Saeki and Patra (2017) Patra et al. (2011) | Inversion (2.8x2.8°) | IEA | ACTM | 32 | NCEP2 (for inversion) | 2003 to 2011 |
| ACTM-CDIAC | // | and forward (1.1x1.1°) | CDIAC | ACTM | 32 | JRA25 for forward | 2003 to 2011 |
| TM5-4DVar | Basu et al. (2013) | 3°x2° | EDGAR +CDIAC | TM5 | 25 | ERA interim | 2007 to 2012 |
| GEOS-Chem | Deng et al. (2014) | 4°x5° | CDIAC, ICOADS and 3-D aviation | GEOS | 47 | GEOS5 | 2009 to 2011 |

**Table 1.** List of the inverse modelling systems used in this study and general characteristics. *with updates documented at https://atmosphere.copernicus.eu/. **with updates documented at http://carbontracker.noaa.gov.

Masarie and Tans 1995; Dlugokencky and Tans 2018). $CO_2$ emissions from land-use-change (ELUC) are the net sum of all anthropogenic activities: deforestation, afforestation, logging and shifting cultivation. Total emissions are estimated, following the bookkeeping method (Houghton, 2003; 2012), with complementary inter-annual variability calculated from satellite data when available (van der Werf et al., 2010; Giglio et al., 2013). The average ELUC for the year 2009 to 2011 included here

5 is estimated to be 0.85 PgC/yr with an uncertainty of 0.5 PgC/yr. These emissions are added to the GCP2016 land sink for comparison to atmospheric inversion estimates.

Finally, the land sink SLAND is estimated in GCP2016 as a residual from all other components of the carbon budget. The GCP2016 method treats the riverine flux of carbon from land to ocean to atmosphere as separate components of the total air-land and air-sea fluxes and subtracts an estimate of this flux (0.45 PgC/yr; Jacobson et al. 2007) from the pCO2-based sea-to-air

10 flux estimates to match estimates of the anthropogenic ocean sink alone. Because the land sink is a residual, this increase in the magnitude of the ocean sink results in a corresponding reduction by 0.45 PgC/yr in the magnitude of the land sink in GCP2016.



To compare to atmospheric inverse flux estimates, which represent the total air-sea and air-land fluxes, we have adjusted the GCP2016 ocean and land flux estimates by this same 0.45 PgC/yr, decreasing the ocean sink and increasing the land sink. Note that we do not show GCP2016 estimates here as a truth metric against which to evaluate the models, but rather as one estimate of an internally consistent global budget that provides a useful reference for exploring axes of variability in our models
and comparing to other community estimates.

### 2.3 HIPPO observations and fitting procedures

The HIPPO project (Wofsy, 2011) used the NSF/NCAR Gulfstream V aircraft (GV) to conduct 5 month-long campaigns in different seasons over three years (2009-2011, see supplement), that consisted of vertical profiling along north-south Pacific
transects between 87°N and 67°S. The 5 campaigns included 9 transects of the NE Pacific. We exclude observations over North America conducted between Colorado and Alaska (Fig. S1). HIPPO flew three different in situ $CO_2$ instruments and 2 whole air samplers with laboratory $CO_2$ measurements. We use the recommended CO2.X variable which comes primarily from the Harvard Quantum Cascade Laser Spectrometer (QCLS), gap filled during calibration sequences, and compare to the other systems to constrain potential systematic biases (see supplement). We calculate the NE vertical gradient as the difference
between the average from the surface to 800 hPa for the lower troposphere (LT), and the average from 800 hPa to 400 hPa for the upper troposphere (UT), spanning the latitude range from 20°N to 90°N. To do this, we first detrend the observations and model flight-track output by subtracting a deseasonalized and smoothed long-term trend record from Mauna Loa, and bin the observations by 100 hPa in pressure and 5 degrees in latitude bins. We then fit each bin with a 2-harmonic curve and constant offset, and average the resulting fits across boxes and pressure levels, with latitude weighting (see supplement). Fig. 1
shows the resulting daily fit of the annual cycle for the HIPPO observations and model simulations of the NE vertical gradient. Qualitatively, it shows that most models reproduce the $CO_2$ cycle well, with positive gradients in winter over a broad peak and negative gradients in summer over a narrower trough. The three CarbonTracker inversions (CT2016, CTE2016-FT, and CTE2017-FT) have somewhat lower seasonal gradient amplitude, while the two ACTM inversions (ACTM-IEA and ACTM-CDIAC) show larger amplitude. More quantitative details are given in Section 3.1. To illustrate the temporal coverage of the
observations, we plot the measurements of the 9 HIPPO transects on Fig. 1 as simple differences of the latitude weighted average concentrations within the LT and UT boxes for each transect, while an example of a fit to an individual bin is shown in Fig. S1.

The QCLS instrument has a very high precision (Santoni et al., 2014), and for all 5 $CO_2$ systems on the GV the instrumental precision is negligible for the large-scale average metrics we present here. More relevant sources of uncertainty are associated
with the potential for altitude dependent biases that might result from inlet or cabin-pressure effects, and misrepresentation of synoptic transport in the models. We estimate uncertainty in the annual-mean NE vertical gradient metric by comparison of the 5 independent instruments and whole air samplers to be ± 0.15 ppm (see supplement), and uncertainty on the individual HIPPO transect values to range from 0.02 ppm to 0.48 ppm as shown by the vertical bars in Fig. 1 These values are derived from the maximum absolute differences between the sensors, which we conservatively treat as best-guess 1 $\sigma$ uncertainty estimates.



These uncertainty estimates correspond to the vertical gradient as observed by the HIPPO flight tracks and calculated with the fitting procedure used here. Because we use model output along the flight tracks and treat model output and observations identically in our calculations, we do not include an estimate of potential spatial sampling bias, but we do use model output to assess the spatial representativeness of our calculated metrics with respect to full curtain averages and full zonal means in the supplement. Also, because the models are driven by reanalysis winds, they should capture the position of synoptic systems and associated transport. However, the wind fields and model transport may be biased which could result in different vertical gradients for reasons unrelated to the fluxes of interest. We have estimated synoptic variability in the vertical gradient metric and find a worst-case potential model synoptic sampling bias of $\pm$ 0.06 ppm for the annual mean, $\pm$ 0.14 ppm for JFM, $\pm$ 0.15 ppm for JAS (1 $\sigma$; see supplement).

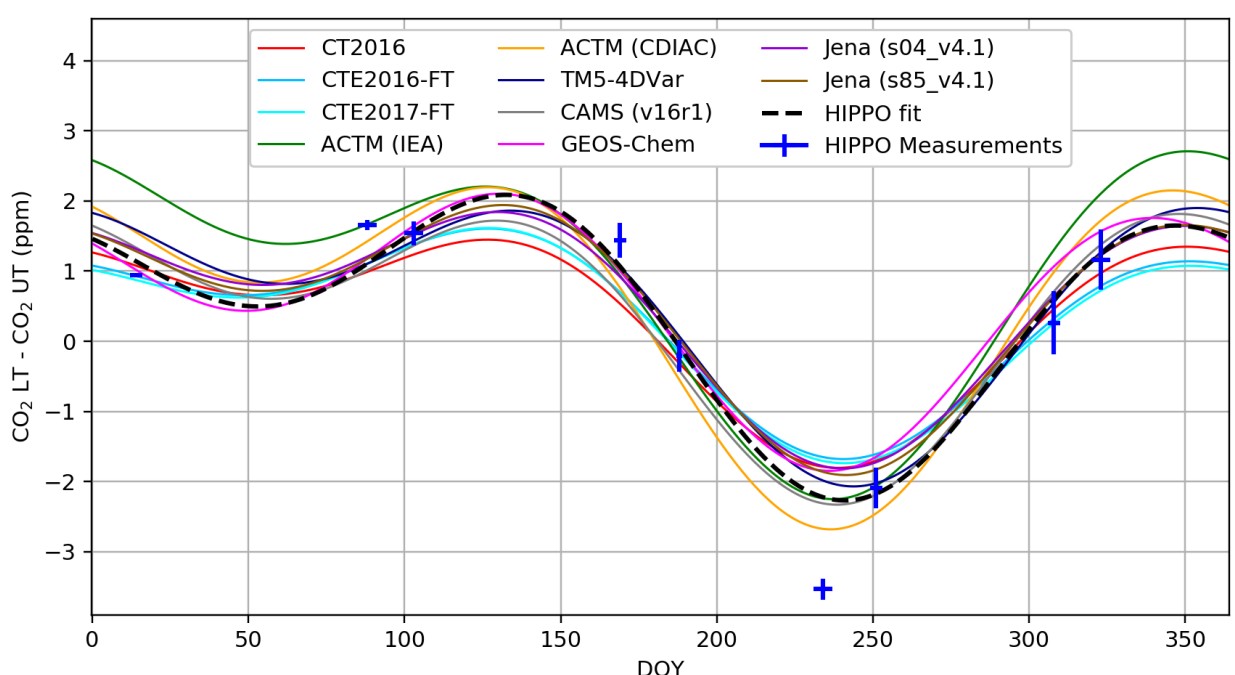

**Figure 1.** Reconstructed annual cycle in northern extratropical vertical $CO_2$ gradients, obtained from 2-harmonic fits of the HIPPO data and correspondingly sampled model outputs, averaged over 20°N to 90°N (surface to 800 hPa minus 800 hPa to 400 hPa). The $CO_2$ average curtain observations for each of 9 atmospheric transects have been added on the graph to illustrate the data uncertainties and temporal coverage, the y-axis errorbar is derived from the range of disagreement among the three in situ instruments on board (QCLS, OMS, and AO2; see supplement), the line average is derived from the CO2.X merged dataset. The horizontal whiskers represent the time span of the flights contributing to each average. The observed line shown here is not a direct fit to the observation points, but rather comes from an average of fits to individual 100 hPa by 5 degree latitude bins as described in the text.



## 3 Results

### 3.1 Fluxes and posterior $CO_2$ comparisons with HIPPO

Each individual inversion system adjusts fluxes to fit the concentration fields with its given transport scheme and a priori source/sink information. Biases can appear in the retrieved posterior $CO_2$ resulting from errors in the estimated fluxes or from
specific biases in transport to the location of the independent data (here in particular vertical transport to the upper atmosphere). We first evaluate if the spread of retrieved land fluxes over different zonal bands is correlated with NE vertical $CO_2$ gradients and if the modelled gradients match observations, as was previously done for the T3L2 models by Stephens et al. (2007). Fig. 2a presents the results for the HIPPO vertical gradients and model fluxes, broken into NE and T + SE regions for the years 2009-2011. The mean and relative spread of 10 simulations for the posterior annual mean NE land flux is 2.24 PgC/yr ± 0.29
PgC/yr (13 %, 1 $\sigma$). Aside from the ACTM-IEA simulation, all models are within the uncertainty range of 0.15 ppm or 50 % of the measured vertical gradient. This contrasts to the TransCom3 Level 2 simulations which had a range of 3.54 PgC/yr ± 1.04 (43 %) PgC/yr for NE land flux and disagreed with the observed vertical gradient by as much as 1.3 ppm (186 %). As listed in Table 1, the inversions have significant differences in transport model, resolution, and driving meteorology and are converging despite these differences. In addition, there are no apparent relationships between vertical gradients and NE nor T + SE land
fluxes. The standard deviation across 10 simulations on the difference between NE Land and T + SE is 0.4 PgC/yr while it was 2.1 PgC/yr in T3L2 (Gurney et al., 2004; Gurney and Denning, 2013) and 1.28 PgC/yr in RECCAP (Peylin et al., 2013), representing a steady and dramatic convergence of model estimates over the past 15 years. We reproduce the Stephens et al. (2007) annual mean figure in Fig. 2b, with the exception of showing T + SE instead of T, to highlight those differences. It is important to note that these results correspond to a different period and different models, with a smaller network of assimilated
in-situ network measurements, and assimilation of monthly-mean rather than discrete measurements. We took advantage of the 2 models that span the 1992-1996 period, CAMS (v16r1) and Jena (s85_v4.1) to further investigate differences from the T3L2 period. Those two models are quite close to the 2009-2011 vertical gradient observations (Fig. 2a), but they both overestimate the 1992-1996 vertical gradients (Fig. 2b). Notably, they fall along the lines fit to the T3L2 models in Fig. 2b, which could be a coincidence, but might also suggest that despite agreeing with the other models on the latitudinal flux distribution for 2009-2011
these models overestimate tropical sources and northern sinks during 1992-1996. This would require that these models be more dependent on vertical mixing biases in the earlier period. The different number of assimilated sites is one potential factor that might explain different biases in retrieved fluxes for these two periods, but this is not seen for the comparison of the 2 versions of the Jena model assimilating different numbers of sites during 2009-2011. It is worth noting that reanalyses of meteorological observations have noticeably improved thanks to the increasing assimilation of satellite data from the mid 1990's (e.g., Gelaro
et al., 2017), and improved transport fields may explain the improved performance with respect to observations for these 2 models during this time. One concern is the spatial representativeness of the HIPPO measurements which were made over the Pacific Ocean while the light aircraft observations used by Stephens et al. (2007) were mostly measuring profiles over land. We discuss this issue in the supplement and show that across models HIPPO vertical gradients are significantly representative of the zonal mean for the three-year mean and every year individually (Fig. S5). Seasonally (Fig. S6), it appears that the vertical





gradients are representative of the parallel 150W for winter (JFM), spring (AMJ) and fall (OND) seasons, representative of the zonal mean for winter (JFM) and fall (OND), and representative of the zonal average over land only in boreal summer (JAS). We did find a significant correlation between vertical gradients defined by the HIPPO flight tracks and land zonal means during summer (JAS), when vertical gradients are weak.

Fig. 2c and Fig. 2d show the vertical gradients and fluxes for 2009-2011 winter (JFM) and summer (JAS). The agreement between the models and HIPPO observations is not as strong as for annual means. The vertical gradient in the NE winter is reasonably well reproduced by 6 models within the uncertainty range of 0.17 ppm and the others with differences lower than 0.5 ppm. The ACTM-IEA inversion is an outlier and overestimates by 1 ppm the winter season average.

There are generally larger differences between observed and modelled vertical gradients in northern hemisphere summer (JAS),
with only 1 model (ACTM-IEA) within observation error bars, but the whole range of values is only 0.75 ppm. In this case a linear relationship ($r^2$=0.4) is found between the modelled vertical gradient and the retrieved T + SE fluxes, but not for the NE flux. Results shown on the Fig. 2d and the Fig. S7 are consistent. There is a significant relationship between HIPPO and the land only zonal average vertical gradient and both are correlated with the T + SE fluxes.

For the annual means and winter there are no statistical relationships between the vertical gradients and the retrieved fluxes.
This suggests that northern hemisphere vertical mixing errors do not play a major role in biasing the flux estimation across these models. However, the retrieved fluxes can still be biased because of the transport errors.

One potential limitation in our analysis could be the use of similar meteorological fields from the ECMWF base analysis and forecast cycle, which is the case for 5 out of 10 simulations. A careful comparison of model transport suggests that nudging to a particular reanalysis product does not imply identical tracer transport between the models (e.g., Prather et al., 2008; Locatelli
et al., 2015; Orbe et al., 2017). The transport errors arise from resolved advection and heavily parameterized transport schemes such as convection and boundary layer mixing (Locatelli et al., 2015; Orbe et al., 2017; Krol et al., 2018). Qualitatively, we cannot distinguish the $CO_2$ vertical gradient from models using ERA-Interim winds from the 5 other models.

## 3.2  The latitudinal distribution retrieved land fluxes

In this section, we present the retrieved land flux partitioning between the NE and the T + SE, as shown in Fig. 3 and on 2. Because the total sink is the sum of T + SE and NE, these lines have a slope of -1 and any deviation perpendicular to the lines indicates disagreement on the total land sink. As noted in the previous section, inverse modelling results for the HIPPO period (2009-2011) are remarkably close to one another (Fig. 3a). These results converge on a NE land sink value slightly larger than 2 PgC/yr (-2.24 ± 0.29 PgC/yr) and a T + SE land sink of -0.38 ± 0.31 PgC/yr. In Fig. 3, multi model means are represented
by blue diamonds and associated error bars are estimated by the standard deviation across models. The 2009-2011 period is marked by a large tropical land sink because of the strong La Niña event of 2011 (Bastos et al., 2013; Poulter et al., 2014). For these 3 years, the models clearly indicate a negative flux over the Tropics and SE land. There are also increasing lines of evidence that deforestation and climate stress over tropics have been moderated in recent decades (e.g., 2000s) compared to the 1990s (Kondo et al., 2018).





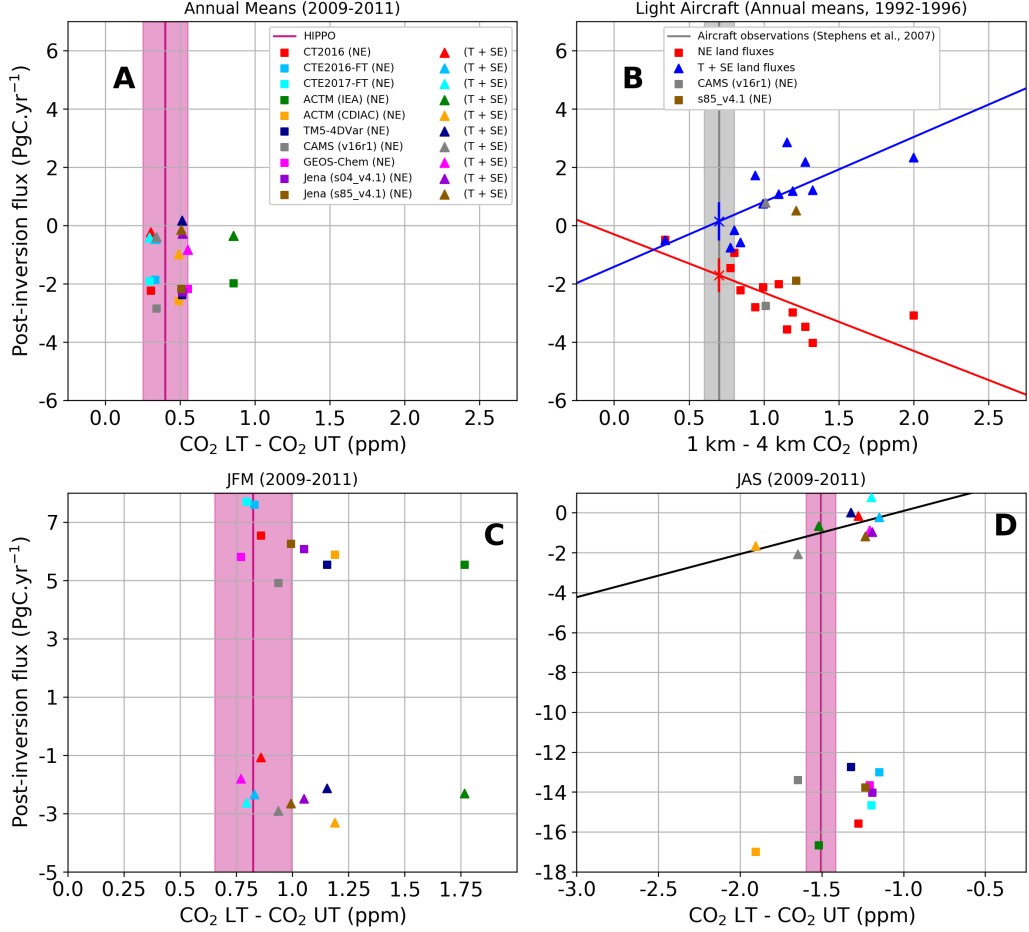

**Figure 2.** Retrieved fluxes versus NE vertical gradients. A) Annual mean NE land and T + SE land fluxes versus posterior NE vertical gradients (lower minus upper troposphere) from model output along HIPPO flight tracks and HIPPO observations (pink line) for the period 2009-2011. The shaded area represents an estimate of measurement uncertainty of $\pm$ 0.15 ppm for the annual mean, as estimated in the Sect. 2 of the supplement. Inverse model posterior concentration gradients and fluxes are shown as points (squares = NE, triangles = T + SE). The vertical axis represents the integrated annual mean land fluxes (PgC/yr). B) Same as (A) but for 1992-1996 and showing TransCom3 Level 2 models and our 2 current models that span this time period, showing dependence of posterior fluxes on transport and a large range of transport biases. Annual mean NE (red squares) and T + SE (blue triangles) land carbon fluxes for 1992-1996 estimated by the 12 T3L2 models plotted as a function of the models' post-inversion predicted mean vertical $CO_2$ gradients at 10 light aircraft profiling sites (adapted from Stephens et al. 2007) with fluxes partitioned by TransCom region. The Jena (s85_v4.1) and the CAMS (v16r1) simulations have also been sampled at the same light aircraft locations but their fluxes are partitioned at 20°N and 20°S. The crosses show our new best estimate of the fluxes estimated by the regression of all T3L2 models. The error bars on these points are estimated using the standard error of the regressions. C) same as panel A for January-February-March (JFM), and D) same as panel A for July-August-September (JAS). For the seasonal plots, the width of the pink bar is of 0.17 ppm for JFM and 0.09 for JAS. On panel D, the black line represents the regression line, shown because the relationship is statistically significant at a 95% confidence interval.



In order to place these recent fluxes estimates in the context of previous studies, we show the flux estimates by the new models that also estimate fluxes for the earlier periods; 2 models have available outputs for the T3L2 period (1992-1996) and 4 for the RECCAP period (2001-2004), as shown in Fig. 3b and 3c. For Jena, one inversion (s85_v4.1) starts in 1985 and is constrained by only 23 atmospheric sites while the other (s04_V4.1) starts in 2004 and uses 59 sites. Interestingly, the difference between

s85_v4.1 and s04_V4.1 for 2009-2011 is rather small (Fig. 3a), less than 0.15 PgC/yr.

According to GCP2016, the total land sink in 2009-2011 was around twice as large (around 3 PgC/yr) than for 1992-1996 (around 1.7 PgC/yr) and 2001-2004 (around 1.3 PgC/yr). This is due to the combined effect of natural inter-annual variability as well as a long-term trend (Ballantyne et al., 2012). The retrieved total land fluxes for all study periods appear to be close to the corresponding GCP estimates with most models falling within the GCP2016 1 $\sigma$ uncertainty range. For the 2001-2004

period, the newer simulations move fluxes parallel to the GCP line in the direction of a weaker tropical source and a weaker NE sink relative to the original RECCAP estimates. For the 1992-1996 period, 1 of the 2 newer simulations shifts fluxes in that same direction, but not as far as suggested by Stephens et al. (2007).

However, we have revisited the Stephens et al. (2007) estimates, by considering the intercept of the regression lines with the aircraft observations rather than the mean of the 3 models nearest the annual mean observations, and evaluating the error using

the standard error of the linear regressions. The selection of 3 models by Stephens et al. (2007) was somewhat arbitrary as they did not directly overlap the observations and did not agree as well as other models seasonally. This new approach relying on the correlated signal from all models leads to a NE flux of -1.7 $\pm$ 0.59 PgC/yr and a T + SE flux of 0.15 $\pm$ 0.66 PgC/yr, a similar shift in NE fluxes but only 2/3 of the shift in T + SE fluxes using the Stephens et al. (2007) subset of models, as shown in Fig. 3b.

For the RECCAP period, we used their Group 1 simulations (JENA, LSCE, MACC-II, CT2011_oi, CTE2013) identified in Peylin et al. (2013), four of which assimilated the observations at the sample time as opposed to using monthly means and all of which solved for fluxes at the resolution of the transport model or for small ecoregions over land. The T + SE flux estimate averaged over the RECCAP Group 1 models is 0.34 $\pm$ 0.27 PgC/yr. This is nearly identical to the average of the new models from this study (0.34 $\pm$ 0.23 PgC/yr; using CTE2016-FT, CTE2017-FT, CT2016, CAMSv16r1, and Jena s85_v4.1). Both

estimate slightly positive T + SE fluxes that are only half of the RECCAP all-model average (0.93 $\pm$ 0.90 PgC/yr). Our NE land sink estimates using newer models are less than the previous estimates in the original T3L2 and RECCAP studies for the 1992-1996 and 2001-2004 periods. Conversely, our new estimates suggest change in the T + SE. flux towards greater uptake / less emission for these periods. Then, to obtain a flux estimate less sensitive to year to year variability we calculate the fluxes for the full 11-year 2004-2014 period (Fig. 3d), for which we have 5 model estimates. For this longer period, the model spread

is largely reduced, in particular for the NE land fluxes, and again we find near neutral T + SE land fluxes. Taking all 4 of the estimation periods together (Table 2) all of our central estimates for T + SE are within 0.4 PgC/yr of zero. The tropical land fluxes are -0.2 $\pm$ 0.3 PgC/yr for 2009-2011 and 0.0 $\pm$ 0.12 PgC/yr for 2004-2014. This implies a consistent uptake of carbon by intact tropical forests over several decades.





**Figure 3.** Tropical and Southern Extratropical (T + SE) versus Northern Extratropical (NE) land fluxes for the periods A) 2009-2011, B) 1992-1996, C) 2001-2004 and D) 2004-2014. The new models used in this study are represented by squares and the average of the available or selected simulations is shown in blue with 1 standard deviation error bars. The pink line and shaded area represents the GCP2016 (river adjusted) estimates of the total land sink for the given period. A) results for the HIPPO period 2009-2011, B) results for the T3L2 period 1992-1996. The TransCom 3 Level 2 outputs (Gurney et al., 2004) are shown in red, with the vertical gradient selected models froms (Stephens et al., 2007) as circles outlined in green and the rest as red squares outlined in black. The intercept of the regression line with the observed vertical gradient (Fig. 2) is use to define our best flux estimate with error bars estimated by the standard error of the linear regression. C) results for the RECCAP period 2001-2004. Also, from Peylin et al. (2013), model means and standard deviations are shown in pink for the subgroup 1 (Jena, LSCEa, MACC-II, CTE2013, CT2011_oi) and in gray for the subgroup 2 (MATCH, CCAM, TrC, NICAM).



| Time period | Source | Number of models | NE Land | T + SE Land |
|---|---|---|---|---|
| 1992-1996 | T3L2 | 12 | -2.42 ± 1.05 | 0.95 ± 1.17 |
| | Stephens et al. 2007 | 3 | -1.52 ± 0.53 | -0.49 ± 0.25 |
| | **T3L2 (Intercept)** | **12** | **-1.70 ± 0.59** | **0.15 ± 0.66** |
| 2001-2004 | | | | |
| | RECCAP All models | 11 | -2.25 ± 0.58 | 0.93 ± 0.90 |
| | RECCAP Group 1 | 5 | -1.85 ± 0.25 | 0.34 ± 0.27 |
| | **This study** | **5** | **-1.67 ± 0.46** | **0.34± 0.27** |
| 2009-2011 | | | | |
| | **This study** | **10** | **-2.24 ± 0.29** | **-0.38 ± 0.31** |
| 2004-2014 | | | | |
| | **This study** | **6** | **-2.17 ± 0.36** | **-0.05 ± 0.11** |

**Table 2.** Previous and our new best estimates (in bold) of the latitudinal partitioning of land fluxes over 4 time periods. All values are in PgC/yr. Values are indicated by the model mean ± 1 standard deviation or 1 $\sigma$ error uncertainties. Regarding the T3L2 period (Gurney et al., 2004), our new estimate for the 1992-1996 period comes from the intercept of the fit lines with the observations in Fig. 2b, and the uncertainties on these values from the standard error on these metrics from the fits. Regarding the RECCAP period (Peylin et al., 2013), our new estimate for the 2001-2004 period is the average of the 5 new models from this study.

### 3.3 Variation in retrieved global carbon budgets

The global carbon budget partitioning for 2009-2011 is shown for our suite of models and for GCP2016 (river adjusted) on Fig. 4 with the model mean and GCP2016 reported in Table 3. On every panel of Fig. 4, the light pink error band shows the constraint imposed by fixing the values to those of GCP2016, and the associated equation is shown on the graph. The pink
diamond represents the GCP2016 estimate while the cross and the gray shaded area show the model mean and 1 standard deviation in darker and 2 standard deviations in lighter gray. For the models, the whole atmosphere growth rate (WAGR) is calculated as the subtraction of the ocean and land sink from the FF emissions. By mass-balance WAGR equals the total surface flux and we use this term to avoid confusion with the atmospheric growth rate as estimated by MBL observations (Dlugokencky and Tans, 2018). GCP2016 uses the observed MBL AGR as an estimate of the WAGR, and provides an uncertainty estimate
associated with this approximation (Le Quéré et al., 2016), with additional discussion in Le Quéré et al. (2018). Note that even though the CAMS results systematically align with the GCP2016 estimates in Fig. 4, the two are independent, except for the FF and for the atmospheric data that serve to estimate the WAGR in GCP2016. By mass balance, the total annual flux must equal the total growth rate integrated over the entire atmosphere, and this is what we refer to as the full AGR.

The integrated ocean versus land fluxes are presented in Fig. 4a. The equation for the range of ocean and land fluxes that would
match FF and WAGR estimates from GCP2016 is also shown on Fig. 4a. The models and GCP2016 agree well on the ocean flux with a mean of -2.04 ± 0.51 PgC/yr for the over the three years 2009-2011. The multi model mean of the land flux is -2.61 ± 0.42 PgC/yr. The GCP2016 land flux is -3.04 ± 0.5 PgC/yr and thus overestimates the model mean. The cloud of model



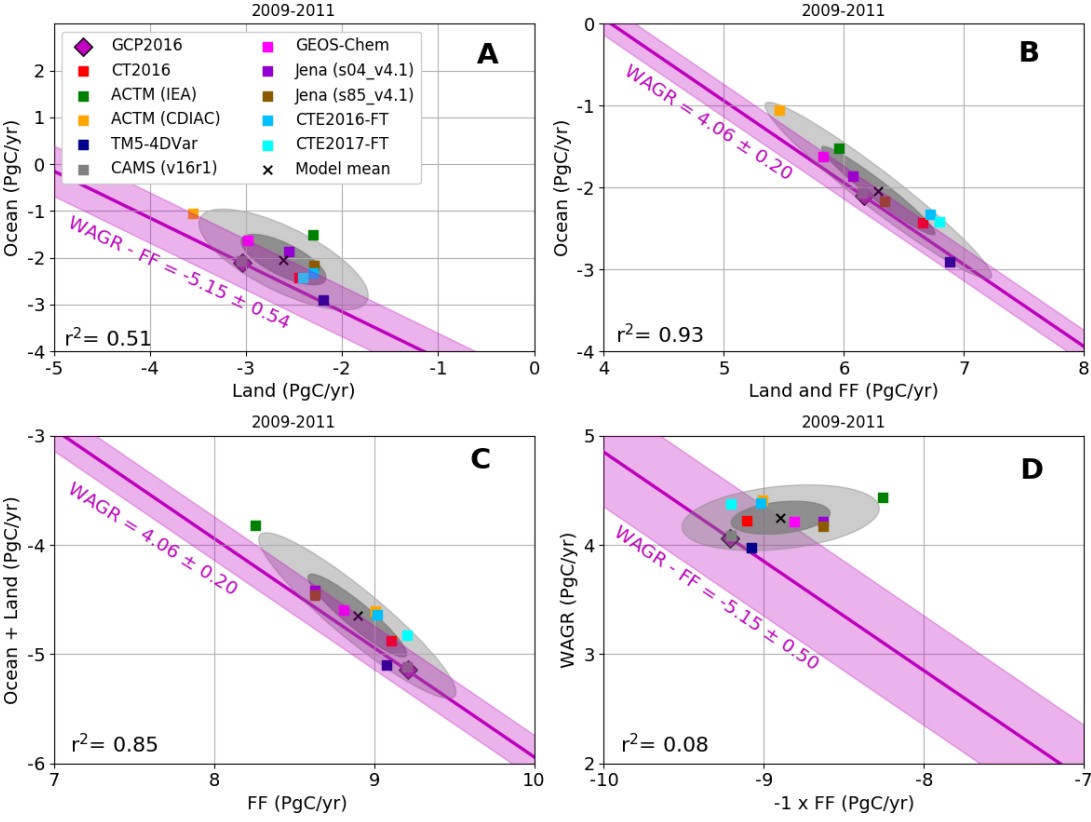

**Figure 4.** Synthesis of globally integrated fluxes for the 2009-2011 period, in PgC/yr. Each inversion is represented by a square and the model mean by a ×. The GCP2016 estimates are a pink diamond, sometimes hard to see because it is superimposed in each panel by the gray CAMS point. we have adjusted the GCP2016 ocean and land flux estimates by the riverine flux of carbon from land to ocean to atmosphere (0.45 PgC/yr; Jacobson et al. 2007; Le Quéré et al. 2018), decreasing the ocean sink and increasing the land sink. The magenta line and light pink shaded area shows the corresponding mass balance estimates from GCP2016. In each panel the line and equation shown represent the sum of the x and y variables, and thus has a slope of -1 and any deviation perpendicular to the line indicates disagreement on the sum. He we use WAGR to refer to the whole atmosphere growth rate, which by mass balance is the total surface flux (see text), and for panels A and D, the WAGR - FF line also equals O + L, while for panels B and C, the WAGR line equals O + L + FF. Ellipses denote the variability around the model means of 1 $\sigma$ (darker gray) and 2 $\sigma$ (lighter gray). A) ocean versus land; B) ocean versus land + FF; C) ocean + land versus FF; D) whole atmosphere growth rate (WAGR) versus -1 × FF.





ocean versus land flux estimates are rather scattered around the model mean with a correlation coefficient of only 0.51. Taking the 2 extreme models the ACTM-CDIAC and TM5-4DVar estimates provide very different distributions of fluxes. Because of an intentionally different FF source, ACTM-CDIAC suggests stronger land sinks, both over the NE and the T + SE regions, and a lower ocean sink while TM5-4DVar suggests the opposite. This leads to a range of around 2 PgC/yr on the model ocean sink.

To better understand the reasons for these discrepancies, and specifically to investigate how much of the land spread in Fig. 4a is a result of differences in fossil fuel priors, we plotted the ocean flux versus the sum of land and FF emissions in Fig. 4b. This figure shows a tight correlation across models for these 2 parameters ($r^2 = 0.93$). Given that prior uncertainties specified in the inversions for ocean fluxes are typically smaller than those for land, and fossil emissions are fixed, this implies for a given ocean and FF flux combination, the models are adjusting the land fluxes while matching $CO_2$ observations. While combining land and fossil fluxes together reduces the random scatter, it does not reduce the range of the continental fluxes, illustrating the fact that models do not simply compensate for biases in fossil priors with land fluxes, but rather that ocean fluxes are affected too (Saeki and Patra, 2017). Conversely, we plot the sum of ocean and land fluxes against FF emissions in Fig. 4c. This figure shows that the ocean + land total sink is largely controlled by the prescribed FF emissions. In general, the models use smaller fossil fuel sources than reported in GCP2016.

Fig. 4d compares the opposite of FF emissions versus the full WAGR, again defined by subtraction of the land and ocean fluxes from FF. The spread in models is not parallel to the line defined by the GCP2016 budget closure. One might expect that models with weaker fossil fuel emissions (to the right in Fig 4d) would retrieve weaker atmospheric growth rates, but this figure shows that generally the opposite is true. We hypothesize that models that overestimate fossil emissions prioritize matching the spatial distribution of $CO_2$ and thus estimate overcompensating sinks. The spatial patterns of the different FF priors must also play a role, as well as the strength of the atmospheric constraint on annual time scales imposed by the inversion systems.

Overall, this analysis suggests that errors in FF priors are larger than the uncertainty prescribed to them, or more specifically, the

|  | FF | Land | Ocean | WAGR |
|---|---|---|---|---|
| GCP2016 | 9.21 ± 0.46 | -3.04 ± 0.50 | -2.05 ± 0.50 | 4.06 ± 0.20 |
| Multi-model | 8.9 ± 0.29 | -2.61 ± 0.42 | -2.04 ± 0.51 | 4.25 ± 0.14 |

**Table 3.** River adjusted Global Carbon budget for 2009 to 2011 inclusive, estimated by the Global Carbon Project 2016 (first row, with river adjusted) and by the suite of models from this study (second row), all values are in PgC/yr. Values are indicated by the model mean ± 1 $\sigma$ error uncertainties, provided by GCP2016 or by the model standard deviation.

range of FF estimates used by leading inversions exceeds the uncertainty that GCP2016 places on the CDIAC estimates. This implies that uncertainties in FF emissions do not adequately consider potential regional biases (Peylin et al., 2011; Thompson et al., 2016; Saeki and Patra, 2017). The large spread of model results along the mass balance line in 4c and the counterintuitive spread away from the mass balance line in 4d both highlight the need to reduce uncertainty in estimates of FF emissions, and to develop modelling systems that relax rigid FF prior constraints and observational systems that can support optimizing FF emission estimates. For the period 1980-2015, the WAGR estimates from GCP2016 are estimated by the MBL AGR of



Dlugokencky and Tans (2018). Only background sites that are located in the MBL are used in this calculation. Ballantyne et al. (2012) calculated a sampling error of 0.38 PgC/yr (2 $\sigma$) among the 40 sites, and GCP2017 estimate uncertainty of $\pm$ 0.19 PgC/yr (1 $\sigma$) for the period 1980-2015 with respect to the WAGR. We show the model retrieved WAGR (equal to total flux) for each individual year on Fig. 5 along with the GCP2016 estimate and error bars. The total spread in WAGR (and total

flux) from the inverse models over the three years 2009-2011 equates to 1.38 PgC as shown in Fig. 5b. This is well outside of the uncertainty range estimated for the extrapolation of MBL measurements, implying several inversions are not rigidly constrained to match observed MBL AGR, even over periods of 3 years. Because $CO_2$ is variably mixed in different years and by different models in the troposphere and between the troposphere and the stratosphere, some inconsistency between the MBL-defined AGR and the total flux of $CO_2$ in the models might be expected. However, using CT2017 as a test case,

the annual difference between the model total surface flux and the observed MBL growth rate over 2000-2016 has a standard deviation of 0.29 PgC/yr, and for three-year averages within this period a standard deviation of only 0.10 PgC/yr, much smaller than the discrepancies shown in Fig. 5. Another potential challenge to inversions having a consistent WAGR during this time may be due to large interannual variability in natural fluxes, with rapid changes resulting from different climatic conditions from the moderate El Niño of 2009 to the strong La Niña of 2011 (Bastos et al., 2013; Poulter et al., 2014). This period has also

been marked by rapid changes in emissions, related to lower emissions in 2009 during the financial crisis and rapid increase in 2010 (Peters et al., 2011). However, Fig. 5 does not indicate that the model WAGR estimates for the years 2009-2011 are more divergent than other years. Further work investigating these differences is needed but beyond the scope of this study. In particular, the length of assimilation window may have an impact. It may also be possible to force the inverse systems to agree, at least within the MBL, with the observationally defined AGR, and this may help to reduce model spread elsewhere.

## 20   4   Summary and future work

Atmospheric transport has long been a major contributor to top-down atmospheric inverse model flux uncertainty. We applied the technique of Stephens et al. (2007) to a suite of state-of-the-art inversion systems assimilating primarily surface observations to take advantage of the unique HIPPO global airborne data set for independent validation in assessing fluxes. We also compared the models to each other and to the GCP2016 carbon budget synthesis. The major findings of these comparisons can

be summarized as follows:

1. Model estimates of the latitudinal distribution of land fluxes are remarkably consistent across models and this represents a convergence over the past 15 years of inverse model development. The standard deviation across our 10 simulations of the difference between northern extratropical land and tropical land fluxes is 0.4 PgC/yr for the period 2009-2011 and 0.43 PgC/yr for the period 2004-2014 across 5 models. These are considerable reductions from 2.1 PgC/yr for 12

simulations in T3L2 (differing only in transport modelling) for the period 1992-1996 and 1.28 PgC/yr for 11 simulations in RECCAP study for the period 2001-2004.

2. Our suite of 10 inversions gives modest NE land uptake of -2.22 $\pm$ 0.27 PgC/yr (1 $\sigma$) and a net T + SE uptake of -0.37 $\pm$ 0.31 PgC/yr for 2009-2011 (-0.2 $\pm$ 0.3 PgC/yr for the tropics only). For 2004-2014, a subset of 6 models gives NE





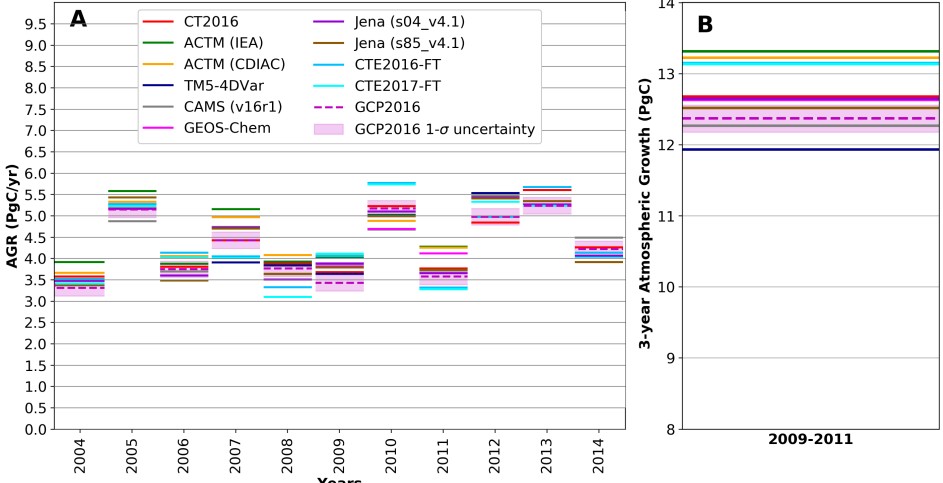

**Figure 5.** Modelled full AGR (lines), equal to total flux, that is the difference between the global FF emissions and the land and ocean fluxes. Atmospheric growth rate from GCP2016, derived from atmospheric $CO_2$ measurements made in the marine boundary layer by the NOAA/ESRL flask network (Masarie and Tans, 1995; Dlugokencky and Tans, 2018) and GCP2016 assigned uncertainty (pink bands).

land uptake of -2.17 ± 0.36 PgC/yr, T + SE uptake of -0.06 ± 0.11 PgC/yr, and T of 0.0 ± 0.12 PgC/yr, thus allowing for deforestation implying a strong uptake in intact tropical forests, in line with forest inventories (Pan et al., 2011).

3. The group of RECCAP models that primarily assimilated discrete rather than monthly mean observations agree with estimates from our subset of 5 newer models regarding the lack of strong net emissions from tropical land. This is not too surprising because most of our models, with the exception of LSCEa, are the updated versions of the same models in the RECCAP Group 1 (Peylin et al., 2013). Those 5 models estimated a net NE land sink of -1.85 ± 0.25 PgC/yr and our subset of 4 models covering the RECCAP period estimate -1.71 ± 0.5 PgC/yr. Regarding T + SE, the newer model estimate is a source of 0.34 ± 0.27 PgC/yr, while it is 0.34 ± 0.27 PgC/yr in RECCAP's Group 1.

4. For the 1992-1996 period, we define an update to the Stephens et al. (2007) result, using the intercept of the model output linear regression with the observed annual mean vertical gradient of 0.7 ppm, leading to a NE land uptake of -1.7 ± 0.57 PgC/yr and a T + SE flux of 0.12 ± 0.62 PgC/yr for 1992-1996. Our results for the more recent decadal period, the 11 years from 2004 to 2014 indicate a somewhat larger NE sink of 2.21 ± 0.34 PgC/yr and a neutral tropical land flux of 0.04 ± 0.13 PgC/yr, in line with a trend of a larger land sink (Sarmiento et al., 2010; Keenan et al., 2016) if shared across both latitudinal bands.

5. We present our best estimates of the latitudinal land flux partitioning for the 4 periods 1992-1996, 2001-2004, 2009-2011, and 2004-2014 in Table 2. These results, and re-evaluations of the T3L2 and RECCAP study results (Table 2)





confirms that the sum of the tropics and southern extratropics have been near neutral for several decades, despite large scale tropical deforestation, and in accordance with a recent review of the tropical land carbon budget (Mitchard, 2018).

6. At global scale, we find in agreement with earlier studies that our model results are strongly dependent on the prescribed FF emissions. While the total of global land and ocean uptake adjusts to match differences in FF emissions, this
compensation is not perfect and counterintuitively models with greater prescribed FF emissions retrieve slower AGR.

7. Our suite of 10 simulations also retrieve surprisingly different three-year full atmospheric growth rates, as defined by the total fluxes. The model range is 1.38 PgC over 3 years, compared to an estimated uncertainty of $\pm$ 0.05 PgC in model matching between MBL $CO_2$ concentration trends and total flux over three years.

Across 7 state-of-the-art systems running 10 inversions, there does not appear to be a correlation between posterior NE vertical
gradients and the retrieved latitudinal distribution of land fluxes in winter and for the annual mean. This is suggesting that northern hemisphere vertical mixing, albeit significant in summer, is not currently the major driver of tropical versus northern extratropical land flux spread. However, transport errors can still contribute significantly to the flux estimates.

Repeating the experiment in T3L2 where transport was the only thing that varied across models would be required to rule out other factors masking a vertical mixing effect, but given the diversity of modern inversion systems this is not practical. Other
components, such as other features of the transport models, the prior fluxes or the inversion method also drive the discrepancies in the global atmospheric budget. The ocean, land, and ocean-land partitioning appear to be a function of the FF prior. This also results in large differences in retrieved AGR or total flux. The fossil fuel and the atmospheric growth rate terms, which are thought to be the best-known terms in the global carbon budget, dominate the model spread at the largest scales and limit our ability to assess the natural fluxes and the ocean-land partitioning from this inversion ensemble.

Our ability to isolate transport effects in this study is limited in comparison to T3L2 in that many other features of the inversion systems also vary; however, this variability allows us to assess the state of the art FF inventories and their importance in the retrieved flux estimates. Gurney et al. (2005), Peylin et al. (2011) and Saeki and Patra (2017) already demonstrated the importance of FF emission uncertainties in inverse modelling studies, suggesting the importance of temporally defined emission inventory. However, Peylin et al. (2011) found that transport errors were still the main source of uncertainty in regional
inversions. With the aim of quantifying $CO_2$ fluxes at regional scales, it is more than ever necessary to assess systematic errors of inverse modelling results with independent in-situ observations. FF emissions could be optimized with the addition of additional species (e.g., Turnbull et al., 2011; Nathan et al., 2018), such as carbon monoxide (Liu et al., 2017; Bowman et al., 2017), although it can be challenging at the most local and urban scales (Ammoura et al., 2016).

There is a significant correlation between NE vertical gradients and the T + SE retrieved fluxes in summer only. This study
reaffirms that systematic evaluation of posterior concentrations against independent measurements is essential to assess the biases and accuracy of inverse modelling systems. Future work will naturally involve comparison against $CO_2$ observations from the more recent NASA Atmospheric Tomography (ATom) project, which is similar to HIPPO, but is augmented with additional flights over the Atlantic Ocean (see e.g. Prather et al. 2017) and an extensive atmospheric chemistry payload, and will involve the inclusion of models assimilating satellite total column $CO_2$ measurements. It is possible that the larger observation



coverage from satellite observations, expanded $^{14}CO_2$ measurements, and urban and power-plant scale observations will help to narrow down the FF emissions, which in turn will allow us to better evaluate inverse model global and regional land and ocean $CO_2$ flux estimates.

*Acknowledgements.* We thank the entire HIPPO team for making these measurements possible and available. We gratefully acknowledge all the hard work that went into collecting and cross-validating the HIPPO $CO_2$ data set: for the Harvard QCLS and OMS instruments by Bruce Daube, Greg Santoni, Jasna Pittman, Rodrigo Jimènez-Pizarro, Roisin Commane, Bin Xiang, and Sunyoung Park; for the NCAR AO2 instrument and NCAR/Scripps Medusa flask sampler by Ralph Keeling, Jonathan Bent, Andrew Watt, and Stephen Shertz; and for the NOAA PFP sampler by Colm Sweeney, Fred Moore, Ben Miller, Steve Montzka, Eric Hintsa, Jim Elkins, and Sonja Wolter. We would like to thank Andrew Jacobson for helping in the analysis of the Carbon Tracker results and the entire CarbonTracker team. CarbonTracker CT2016 and CarbonTracker CT2017 results are provided by NOAA ESRL, Boulder, Colorado, USA from the website at http://carbontracker.noaa.gov. The National Center for Atmospheric Research is sponsored by the National Science Foundation. This work was supported by NASA ROSES grant NNX15AG87G. F. Chevallier is funded by the Copernicus Atmosphere Monitoring Service, implemented by the European Centre for Medium-Range Weather Forecasts (ECMWF) on behalf of the European Commission, and was granted access to the HPC resources of TGCC under the allocation A0030102201. F. Deng was supported by Environment and Climate Change Canada (ECCC) and Canada Space Agency (CSA).



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
