# Peer review of "Global atmospheric CO2 inverse models converging on neutral tropical land exchange, but disagreeing on fossil fuel and atmospheric growth rate"

_Biogeosciences, 2018_

## Referee Comment (RC1) · A.P. Ballantyne (Referee) · 24 Oct 2018

Review: Global atmospheric CO2 inverse models converging on neutral tropical land exchange but diverging on fossil fuel and atmospheric growth rate
Benjamin Gaubert et al.

**Summary:** Here Gaubert et al. use recent meridional flight observations to test the latest itteration of atmospheric inversion models and their ability to capture vertical profiles of atmospheric CO2. There results are somewhat reassuring in that the inverse models with inversed complexity actually appear to becoming more accurate and precise over time, which is not always the case with global scale model development. The authors also arrive at an interesting conclusion that the largest sources of error currently limiting inversion modeling is the spatially explicit fossil fuel inventories and the atmospheric growth rate- two terms that are thought to be well constrained at the global scale. While my expertise are not specifically in inverse modeling, I think that this is an interesting study that highlights the current limitations and advances in inversion approaches and should be published after minor revision. With that in mind I have a couple of ideas that might give this paper impact beyond the inverse modeling community.

**General Comments:**

The global fossil fuel emission inventories only differ by ~10% (Ballantyne, Andres, and Houghton 2015) but it appears that the spatial-temporally explicit inventories differ considerably. It appears that the ACTM (IEA) simulation is the most anomalous among the models considered; however, ACTM (CDIAC) vertical profiles in the northern hemisphere look the most unusual (Fig S2) which seems odd. Is there any way to simply plot the differences in IEA and CDIAC emission estimates to demonstrate how which emission inventory you select for your simulation may greatly alter your results?

Is there anyway to create figure of how NE uptake relative to T+SE uptake has changed over time? This information is nicely compiled in table 2, but it is hard to discern any patterns, such a figure could be very interesting to researchers outside the atmospheric inversion community.

There is interesting recent work on changes in relative forest cover, especially within the the tropics, using LANDSAT data that may be relevant to your results (see (Hansen et al. 2013) and subsequent work.

**Specific Comments:**

P2 L3 You may want to specify in the final sentence that it is the regional fossil fuel emission estimates that dominate the uncertainty.

P3 L21 You say what Frankenberg did, but what did Frankenberg discover?  This seems particularly relevant because of the conclusion about the growth rate uncertainty. If we had a gridded XCO2 growth rate for every grid cell on Earth would this improve inversion estimates of regional fluxes?  It is my understanding from work by Houweling that the XCO2 GOSAT measurements do not improve the inversions that much.

P3 L25 What are the 'a priori assumptions' you are referring to?

P3 L30 change to 'global carbon budget estimates (LeQuere et al. 2016)'  You also cite LeQuere 2016 and 2018 is this intentional?

P5 L9 Is this a riverine DIC flux to the ocean? This 0.45 PgC/yr is much less than the recent estimate of 2.1 PgC/yr emissions from inland waters (Raymond et al. 2013).

P6 L14 How much does this gradient vary from the tropics to high latitudes- is this just the NE gradient?

P6 L 28 You might want to present the instrument precision first and then let the reader decide if it is 'negligible'.

Figure 1.  Avoid acronyms in figure captions if you can Day of Year instead of DOY. Also define 'LT' and 'UT' in caption.  Do the HIPPO measurements here represent averages across many years for the same day, or is this just for a single years observations.

P8 L9 - 11 It seems that you are comparing mean from present study to range from Transcom

Figure 2 report actual statistics of relationship in D and let reader decide how significant it is. Use lower case letters on panels for consistency with text.

P9 L12 How are they consistent? Explain. It looks like the slope of the relationship changes in S7 in the land only figure. Is this important?

P9 L25 and in Figs. 3 and 2

P9 L33 See work by Hansen et al. showing no net change in tropical forest cover because decrease in S American deforestation compensated by increased SE Asia deforestation.

P11 L27 'Conversely…' Is this a complete sentence and it is hard to tell any trend from Table 2.

P16 L32 seems like considerable NE land uptake and not 'modest'

P17 L5 What is LSCEa? Is this CAMS?

P17 L8 the same as RECCAP Group 1

P18 L 1 Once again see work by Hansen on changes in forest cover, especially in the tropics.

P18 L 14 'is not practical' really maybe this is opportunity for suggestions on the next inversion MIP.

P18 L 18 see uptake uncertainties by Ballantyne et al. 2015 and change ' limit our ability to assess the natural fluxes at regional scales…'

**References:**

Ballantyne, A. P., R. Andres, and R. Houghton. 2015. "Audit of the Global Carbon Budget: Estimate Errors and Their Impact on Uptake Uncertainty." http://www.biogeosciences.net/12/2565/2015/bg-12-2565-2015.pdf.

Hansen, M. C., P. V. Potapov, R. Moore, M. Hancher, S. A. Turubanova, A. Tyukavina, D. Thau, et al. 2013. "High-Resolution Global Maps of 21st-Century Forest Cover Change." *Science* 342 (6160): 850–53.

Raymond, Peter A., Jens Hartmann, Ronny Lauerwald, Sebastian Sobek, Cory McDonald, Mark Hoover, David Butman, et al. 2013. "Global Carbon Dioxide Emissions from Inland Waters." *Nature* 503 (7476): 355–59.

---

## Short Comment (SC1) · 24 Oct 2018

The comment was uploaded in the form of a supplement:
https://www.biogeosciences-discuss.net/bg-2018-384/bg-2018-384-SC1-supplement.pdf
* * *

---

## Referee Comment (RC2) · R. Law (Referee) · 29 Oct 2018

General comments

This paper uses the HIPPO dataset to evaluate recent atmospheric CO2 inversions and compares the spread of fluxes from the inversions with earlier inversion results and the Global Carbon Project budget. It is a useful presentation of the current state of inversions but I wonder whether more could have been done to increase our understanding of why the current inversions are more convergent than the earlier ones. I also

wonder whether more information could have been extracted from the comparison with the HIPPO data. While these two points could be explored extensively, clearly some choices will have to be made as to what should be added to this paper and what should just be highlighted as areas for potential future work. I will try and provide some more specific suggestions below.

Specific comments

Use of HIPPO data: the analysis presented in the paper effectively reduces the HIPPO data to 3 numbers, the 2009-2011 annual, JFM and JAS mean northern extratropical vertical gradient, for comparison with equivalently sampled model output. I suspect this allows model errors to cancel out in the averaging and makes it harder for the HIPPO data to discriminate between inversions. This is seen even in the seasonal results; the one inversion that matches the JAS gradient is also the one inversion that does not match the annual mean gradient. This raises the question around what value is put on matching the annual mean if seasonal means are not correct. This should at least be more fully discussed in the paper.

Impact of harmonic fitting: Fig S1 shows observations that are not well fitted by the harmonic, and this flows through into Fig 1 where the average HIPPO fit can be a long way from the mean observation point. Is this also the case for the model samples? If they were plotted on Fig 1 or Fig S1, would they scatter around the HIPPO observation point (e.g. the low point at day ∼240) or would they scatter around the fitted lines? Is there information in the outliers to the fit which the models are able to capture? It could be useful to create scatter plots of each binned HIPPO gradient against each equivalent model generated gradient. Do they cluster on a one-to-one line and with what correlation? Do different models give different scatter patterns/correlations? If these figures are informative, perhaps they could be added to the supplementary information. My main concern is whether fitting the harmonic is minimising differences between the models and making it harder to discriminate between them? The low sensitivity to synoptic modelling bias presented in the Supplement would tend to confirm this. Would

Fig 2a look different if instead of taking the annual mean from the harmonic fit, you just averaged the 9 data points - it wouldn't necessarily be representative of an annual mean but as long as the observations and models were treated in the same way that probably isn't important.

Convergence of current inversions compared to previous ones: it would be good to provide some additional discussion in the paper as to why the current inversions are likely to be more convergent in their land estimates than the T3L2 ones. Since the T3L2 inversions all used the same method and priors, it would suggest that either there has been convergence in the transport models or that current methods are less sensitive to the atmospheric data and are more constrained by their priors. This seems possible if current methods are all using data at observed times rather than monthly means and not solving for large regions. This information is not currently in the paper. It would be good to add it (at least to supplementary material) i.e. the inversion method used, and some indication of the number of flask/in-situ atmospheric sites used and how they are used e.g. at measurement time, with what selection in the in-situ case. It would also be helpful to know what the magnitude of the prior land and ocean fluxes are for 2009-2011. This could either be provided as part of the model information in the supplement or perhaps plotted similarly to Fig 3a and Fig 4a. Knowledge of the prior may be particularly useful for understanding the land-ocean partition as I'm guessing it may contribute to some of the inversion differences.

Fossil emissions and atmospheric growth rate: I'm not sure I agree with some of the discussion around Fig. 4d. For example, the sentence on p15, line 17 starting 'One might expect ...' and 'counterintuively' (p18, line 5). The intent of an atmospheric inversion is to constrain fluxes by the atmospheric data. Thus I would expect the AGR to be well fitted and any difference in fossil between inversions to be compensated by variations in the other fluxes. This is what you see in Fig 4c. Hence a near horizontal line (as in Fig 4d.) for the modelled WAGR is what I'd expect from an inversion perspective. I don't think the suggestion of an opposite relationship (p15, line 19 'generally

the opposite is true', also p18 line 5 'lower AGR') is strong enough to make this point, rather that the WAGR is insensitive to the FF estimate used. If we were looking at different timeslices with real variation in fossil emissions (rather than just a variety of FF estimates for the same period) then it might be reasonable to expect a relationship between FF and WAGR but not when we are looking at a fixed period (2009-2011).

In general I found the 'WAGR', 'full AGR' and 'AGR' terminology confusing. Perhaps it is best to just use 'total flux' instead of WAGR since this is what you actually use from the inversion. Alternatively, it may be reasonably easy to calculate the MBL AGR from the inversions since I'd expect each inversion uses these sites and would likely keep information about their posterior fit to these sites. This measure could then be used as a direct comparison to the GCP calculated AGR and would be a useful additional indication of inversion spread.

Given my comments above, I think you need to be more careful with some of the messages that you draw out of the paper in the title, abstract and summary sections. Specifically: (a) 'diverging on fossil fuel and atmospheric growth rate' as used in the title - diverging compared to what? The converging vs diverging language implies that the fossil and AGR terms now have larger uncertainty than the land term. Using the numbers in Table 2 and Table 3, it would be fair to say that choices about fossil prior now have as large an uncertainty as those retrieved for tropical land from the inversions, but both are still larger than that for WAGR. Perhaps this is the main point to be made - that reductions in the inversion spread for land and ocean fluxes, now means that more care needs to be taken with how the fossil term is included in the inversion. Likewise in the abstract, p2, line 3, I don't agree that fossil and AGR terms 'dominate' the model spread since their uncertainties in Table 3 are smaller than those for land and ocean. Perhaps 'contribute to model spread at the largest scales and thus our ability to assess ...' Similarly at p18, line 18. (b) The abstract makes a general statement about agreement with HIPPO (p1, line 9) but the paper only really presents the northern vertical gradient so perhaps this statement should be modified to only include the 'in

particular ...' phrase (unless any of the analysis suggested above provides further insight into the agreement with HIPPO data across inversions) (c) Abstract, p1, line 15. I am not convinced that the large disagreements on ocean-land partitioning are strongly influenced by the prescribed fossil fuel and associated differences in retrieved AGR.

Technical comments

p1, line 6: 'latitudinal distribution' - is this the best term to use since the paper really only focusses on the split between north and tropics+south.

p2, line 19: I found the 'NE' and 'SE' acronyms distracting as I had to stop myself from reading them as North-East and South-East. Perhaps just use 'N/S' or 'NEx/SEx'

p3, line 34: The GCP2016 acronym is used here but not defined until p4.

p4, line 9: missing ')' after Boden et al 2016.

p4, line 10: You note the possibility of using the ACTM inversions to assess the sensitivity to the FF, but it wasn't obvious to me that you actually do that within the analysis presented in the paper.

p6, line 20: It would be good to note somewhere here that the model is sampled along flight tracks i.e. move the comment from p7, line 2-3 earlier.

p7, line 4: suggest replace 'curtain averages' with '150 W transect'

p7, line 5: Add '(Section 4)' after 'supplement'

p8, line 8: perhaps add 'and model' before 'vertical gradients'

p8, line 9: Should this be -2.24?

p8, line 11: It's not clear to me what number is being quoted here. A range of 3.54 seems to match the Fig 2b, but what does the 1.04 refer to? It seems unusual to quote an uncertainty on a range. Perhaps just give the mean and 1-sigma as this would then

be comparable with the number given in line 9.

p8, line 12: 1.3 ppm - this is from one outlier, perhaps better to give the typical mis-match ~0.5 ppm.

p9. line 12: Perhaps 'Results shown in Fig. 2d and Fig. S7 are consistent.'

p9, line 24: Insert 'of' before 'retrieved land fluxes'

p9, line 25: what did you mean by 'and on 2.'?

p9, line 27: Add 'from the GCP2016 estimate' after 'disagreement on the total land sink'

Figure 3 caption: Is this complete? There was no specific information about panel D.

Table 2: The T+SE land flux is the same (0.34 +/- 0.27) for RECCAP Group 1 and This Study. Is this correct?

p13, line 13: 'full AGR'. If this is intended to be the same as the WAGR, then just use WAGR.

p14, Figure 4 caption. line 3 'We' not 'we'. line 6 'Here' not 'He'. I was confused by the use of 'WAGR-FF line' and 'WAGR line' in line 8. If these are for GCP, are they from AGR not WAGR?

p15, line 3: 'Because of the intentionally different FF source'. I don't think this is the explanation for the difference between ACTM-CDIAC and TM5-4DVar, since their FF values seem relatively similar (~9 PgC/y). Perhaps the different prior ocean flux used makes a contribution.

p15, line 16: 'again defined' - perhaps qualify this as just for the models, assuming that the GCP line is from the MBL AGR.

p15, line 25-26: 'counterintuitive spread away from the mass balance line in Fig 4d' - does this require the assumption of a constant airborne fraction? If so, it might be good

to note this.

Figure 5 is difficult to read. Perhaps plot across a smaller range, or plot as differences from GCP. I assume that panel B is basically the same information as the y axis of Fig 4d just as a sum over 3 years rather than the mean. CAMS looks slightly smaller than GCP in this figure but slightly larger in Fig 4d. Panel B is not described in the Figure caption. Replace 'full AGR' with 'WAGR' in figure caption.

p18, line 5: 'lower' instead of 'slower' - though I'm not sure the signal is strong enough to really make this point - ellipse on Fig 4d is close to horizontal.

p18, line 7: not sure where the +/- 0.05 number comes from and exactly what it means.

Supplementary material

p1: define STL

p1: how different are the MLO reference trends subtracted from the models? I would assume they are quite similar since the inversions would do a good job at fitting the MLO data, but it might be worth mentioning.

p1: You might like to add a comment interpreting the results presented in Fig S2 and S3.

p2: The text says your averaging was for 5 degree bins but the example given seems to be for a 10 degree bin. Since most of your analysis only uses 20-87N and below 400 hPa perhaps it would be simpler to only plot these regions in panel 1 and 3.

p3 and p4: In Fig S2 and S3 it would be helpful to put all the CT cases in one row, and put the ACTM cases next to each other. I thought the ACTM cases looked surprisingly different in Fig S2.

Inversion descriptions: Jena, biosphere and fires: 'Constant' - do these priors include a mean seasonal cycle?

---

## Author Comment (AC2) · 3 Dec 2018

The comment was uploaded in the form of a supplement: https://www.biogeosciences-discuss.net/bg-2018-384/bg-2018-384-AC2-supplement.pdf

---

## Author Comment (AC3) · 3 Dec 2018

The comment was uploaded in the form of a supplement:
https://www.biogeosciences-discuss.net/bg-2018-384/bg-2018-384-AC3-supplement.pdf

---

## Author Comment (AC4) · 3 Dec 2018

The comment was uploaded in the form of a supplement:
https://www.biogeosciences-discuss.net/bg-2018-384/bg-2018-384-AC4-supplement.pdf

---

## Author Response (AR1)

We thank both reviewers for their time and helpful comments on ways to improve the paper. Below are our responses to individual comments (original comments in black plain text, *responses in blue italic*).

**Review 1:**

**Summary:** Here Gaubert et al. use recent meridional flight observations to test the latest iteration of atmospheric inversion models and their ability to capture vertical profiles of atmospheric CO2. There results are somewhat reassuring in that the inverse models with inversed complexity actually appear to becoming more accurate and precise over time, which is not always the case with global scale model development. The authors also arrive at an interesting conclusion that the largest sources of error currently limiting inversion modeling is the spatially explicit fossil fuel inventories and the atmospheric growth rate- two terms that are thought to be well constrained at the global Scale. While my expertise are not specifically in inverse modeling, I think that this is an interesting study that highlights the current limitations and advances in inversion approaches and should be published after minor revision. With that in mind I have a couple of ideas that might give this paper impact beyond the inverse modeling community.

**General Comments**

The global fossil fuel emission inventories only differ by ~10% (Ballantyne, Andres, and Houghton 2015) but it appears that the spatial-temporally explicit inventories differ considerably. It appears that the ACTM (IEA) simulation is the most anomalous among the models considered; however, ACTM (CDIAC) vertical profiles in the northern hemisphere look the most unusual (Fig S2) which seems odd. Is there any way to simply plot the differences in IEA and CDIAC emission estimates to demonstrate how which emission inventory you select for your simulation may greatly alter your results?

**Authors' response:** *We agree that this relationship is counterintuitive and requires extra care to communicate. We have added text on page 9 of "For ACTM, the global annual IEA emissions are less than CDIAC (Figure 4C and D) which results in a weaker northern extratropical sink (Figure 2A and 3A) which corresponds with a more positive LT-UT northern extratropical gradient (Figures 2A and S2) and a more positive N-S gradient (Figure S2), comparing just the 2 ACTM versions. Differences across inversion systems in Figure S2 also depend on transport and inversion scheme and the resulting spatial distribution of sources and sinks."*
*The detailed differences between these 2 emission estimates, and their impact on the 2 inversions are presented in Saeki and Patra (2017), which is also cited in the main text.*

Is there anyway to create figure of how NE uptake relative to T+SE uptake has changed over time? This information is nicely compiled in table 2, but it is hard to discern any patterns, such a figure could be very interesting to researchers outside the atmospheric inversion community. There is interesting recent work on changes in relative forest cover, especially within the tropics, using LANDSAT data that may be relevant to your results (see (Hansen et al. 2013) and subsequent work.

**Authors' response:** *Good suggestion. The time series of the NET uptake and the T+SET is now presented in a new figure in the manuscript. The 2004-2014 tropical land sink is 0.71 PgC/yr greater than 1994-2004. The LULUC (primarily tropical) estimated in GCP2016 decreased from 1.33PgC/yr to 0.94PgC/yr, so this implies an increase in uptake in intact tropical forests.*
*In the conclusion, we added the following:*
*"We present in Fig. 6 the time series of the NET and T+SET land fluxes from 1979 to 2016, using all simulations available in this study. This figure shows a decrease of the T+SET land flux by 0.71 PgC/yr, from +0.56 PgC/yr to -0.15 PgC/yr between the decades 1994-2004 and 2004-2014, respectively. The land-use change flux estimated by GCP2017 was nearly identical for these two time periods (+1.31 and +1.29 PgC/yr, respectively), and assuming these numbers primarily reflect tropical land-use change emissions this implies an increase in the intact tropical forest sink on decadal timescales. Our re-evaluations of the T3L2 and RECCAP study results (Table 2) confirms that the sum of the tropics and southern extratropics have been near neutral for several decades, despite large scale tropical deforestation, and in accordance with the recent literature on the tropical land carbon budget (Hansen et al., 2013; Keenan et al., 2016; Mitchard, 2018)."*

[Figure]

**Figure 6:** Time series of annual land fluxes for the NET (top panel) and the T+SET (bottom panel). The black line represents the model mean and standard deviation derived from available simulations, the number of simulations is shown by the numbers below the curve. The standard deviation is shown only if there are more than 2 model simulations available. Estimates from the specific period (Table 2) are added as multi-year average and standard deviation (shaded area).

**Specific Comments:**

P2 L3 You may want to specify in the final sentence that it is the regional fossil fuel emission estimates that dominate the uncertainty.

**Authors' response:** *Our analyses do not support specific conclusions regarding regional uncertainties in the fossil fuel emission inventories. But we have changed the sentence from*
*"but we show that they dominate the model spread at the largest scales and currently limit our ability to assess regional scale terrestrial fluxes and ocean-land partitioning from the model ensemble." To:*
*"but we show that they currently limit our ability to assess regional scale terrestrial fluxes and ocean-land partitioning from the model ensemble."*

P3 L21 You say what Frankenberg did, but what did Frankenberg discover? This seems particularly relevant because of the conclusion about the growth rate uncertainty. If we had a gridded XCO2 growth rate for every grid cell on Earth would this improve inversion estimates of regional fluxes? It is my understanding from work by Houweling that the XCO2 GOSAT measurements do not improve the inversions that much.

**Authors' response:** *We added the following sentence: "They found that despite an overall agreement between inversions and HIPPO measurements, systematic model transport errors remain important."*
*There is also a discussion on the following page that: "Those inverse exercises, however, are still sensitive to satellite retrieval algorithms and the inversions' prior assumptions. In particular, the results are sensitive to systematic errors from transport and satellite retrievals (Houweling et al., 2010, 2015; Chevallier, 2015)."*
*This is confirming the role of systematic errors, which are detrimental to inverse modeling, having a gridded $XCO_2$ growth rate would be useful only in the case it is not biased. Otherwise a bias in satellite observations would led to a bias in the retrieved fluxes.*

P3 L25 What are the 'a priori assumptions' you are referring to?

**Authors' response:** *We replaced the "a priori assumptions" by "and on prior flux estimates and their uncertainties used in the inversions."*

P3 L30 change to 'global carbon budget estimates (LeQuere et al. 2016)' You also cite LeQuere 2016 and 2018 is this intentional?

**Authors' response:** *It is intentional, this aspect is discussed in Sect. 2.2, where both GCP 2016 and GCP 2017 are presented.*
*We updated the sentence there to introduce 'our acronym': "Second, we compare the observationally-constrained fluxes for latitudinal bands and on a global scale both across models and to budget estimates provided by the Global Carbon budget 2016 (hereafter denoted GCP2016; Le Quéré et al. 2016)."*
*Here is the text in the appropriate section:*

*"The Global Carbon Project (GCP) gathers observational and model-based flux estimates from multiple organizations and research groups around the world to yearly report a global budget of atmospheric $CO_2$ (Le Quéré et al., 2016). The GCP Carbon Budget version 2016 (Le Quéré et al., 2016) is the most recent version with flux estimates forced to balance globally. The most recent version (GCP2017; Le Quéré et al. 2018) separated an explicit unknown ocean or land flux term, which prevents simple comparisons of the type presented here. Specifically, …"*

P5 L9 Is this a riverine DIC flux to the ocean? This 0.45 PgC/yr is much less than the recent estimate of 2.1 PgC/yr emissions from inland waters (Raymond et al. 2013).

**Authors' response:** *Yes, it is a flux to the ocean, but as such it is not comparable to the emission from inland waters, which is a different term in the budget. We recognize there is significant uncertainty in the river-to-ocean flux, but for consistency with GCP2016 we present their results using their recommended value, per Table 7 subscript in Le Quéré et al. (2016): "b Estimates are not corrected for the influence of river fluxes, which would reduce the fluxes by 0.45 GtC/yr." If this value were larger, as suggested by Resplandy et al. (2018) then the magenta diamond in Figures 2A and 2B would shift down and to the right, but the position of the GCP lines would not change.*

P6 L14 How much does this gradient vary from the tropics to high latitudes- is this just the NE gradient?

**Authors' response:** *Figure S2 and Figure S3 show the $CO_2$ annual mean and seasonal amplitude, which show that lower tropospheric $CO_2$ is clearly enhanced for latitudes north of 20 N. The gradient we present is just for the NET (spanning the range of latitude, from 20 N to 90 N).*

P6 L 28 You might want to present the instrument precision first and then let the reader decide if it is 'negligible'.

**Authors' response:** *We updated the sentence to: "The QCLS instrument has a 1-σ precision of 20 ppb (Santoni et al., 2014),"*

Figure 1. Avoid acronyms in figure captions if you can Day of Year instead of DOY.
Also define 'LT' and 'UT' in caption.

**Authors' response:** *This has been corrected.*

Do the HIPPO measurements here represent averages across many years for the same day, or is this just for a single years observations.

**Authors' response:** *The HIPPO measurements spanned 3 years but in 5 campaigns with no overlap in time of year. The fits to HIPPO represent estimates of a climatological year.*

P8 L9 - 11 It seems that you are comparing mean from present study to range from Transcom.

**Authors' response:** *We changed this to: "This contrasts with the TransCom3 Level 2 simulations which had an annual mean of 2.42 PgC/yr +/- 1.05 (43 %) PgC/yr"*

Figure 2 report actual statistics of relationship in D and let reader decide how significant it is. Use lower case letters on panels for consistency with text.

**Authors' response:** *To be consistent, we now use upper case letters in the text. We plotted the black line on Figure D because the statistical relationship is significant at 95 %. In other words, the p-value is less than 0.05, so we reject the null hypothesis. A statistical significance is an objective criterion.*

P9 L12 How are they consistent? Explain. It looks like the slope of the relationship changes in S7 in the land only figure. Is this important?

**Authors' response:** *We have deleted the sentence "Results shown in Fig. 2D and the Fig. S7 are consistent," as it was not clear and the main point is in the following sentence. We have also added ", but with a slope of 2.16 ppm / PgC/yr for HIPPO while it is 0.93 ppm / PgC/yr over land where the vertical gradients are bigger."*

P9 L25 and in Figs. 3 and 2

**Authors' response:** *It has been corrected to "and in Fig. 3 and Fig. 2" thanks.*

P9 L33 See work by Hansen et al. showing no net change in tropical forest cover because decrease in S American deforestation compensated by increased SE Asia deforestation.

**Authors' response:** *We updated the sentence to: "There are also increasing lines of evidence that the rate of deforestation and climate stress over tropics have been moderated in recent decades (e.g., 2000s), compared to the 1990s (Kondo et al., 2018), with a reduced change in tropical forest cover because the decrease in the south American deforestation has been compensated by an increased south east Asian deforestation (Hansen et al., 2013)."*

P11 L27 'Conversely…' Is this a complete sentence and it is hard to tell any trend from Table 2.

**Authors' response:** *We changed the sentence to "Conversely, our new estimates suggest a change in the T + SET flux towards greater uptake / less emission for these periods; we found a decrease of the T + SET land flux by 0.71 PgC/yr, from 0.56 ± 0.32 PgC/yr for the 1994-2004 period compare to -0.15 ± 0.43 PgC/yr for the 2004-2014 period (Figure S9)."*

P16 L32 seems like considerable NE land uptake and not 'modest'

**Authors' response:** *We changed the sentence to "Our suite of 10 inversions gives a NE land uptake of …"*

P17 L5 What is LSCEa? Is this CAMS?

**Authors' response:** *The LSCEa acronym has been used in Peylin et al. 2013 (see their Table 1), it is an inversion system developed at LSCE, that actually differs from the MACC-II (which evolved to CAMS) system also used in the same study.*

P17 L8 the same as RECCAP Group 1

**Authors' response:** *Yes, both end up having the same value, note that the standard deviation is 0.27.*

P18 L 1 Once again see work by Hansen on changes in forest cover, especially in the tropics.

**Authors' response:** *The updated paragraph now reads:*
*"We present in Fig. 6 the time series of the NET and T+SET land fluxes from 1979 to 2016, using all simulations available in this study. This figure shows a decrease of the T+SET land flux by 0.71 PgC/yr, from +0.56 PgC/yr to -0.15 PgC/yr between the decades 1994-2004 and 2004-2014, respectively. The land-use change flux estimated by GCP2017 was nearly identical for these two time periods (+1.31 and +1.29 PgC/yr, respectively), and assuming these numbers primarily reflect tropical land-use change emissions this implies an increase in the intact tropical forest sink on decadal timescales. Our re-evaluations of the T3L2 and RECCAP study results (Table 2) confirms that the sum of the tropics and southern extratropics have been near neutral for several decades, despite large scale tropical deforestation, and in accordance with the recent literature on the tropical land carbon budget (Hansen et al., 2013; Keenan et al., 2016; Mitchard, 2018)."*

P18 L14 'is not practical' really maybe this is opportunity for suggestions on the next inversion MIP.

**Authors' response:** *We add this sentence: "Having a common FF prior, eventually also optimized and with known uncertainties, would improve our ability to retrieve the natural fluxes."*

P18 L 18 see uptake uncertainties by Ballantyne et al. 2015 and change ' limit our ability to assess the natural fluxes at regional scales…'

**Authors' response:** *We updated the sentence to: "The increase of the absolute error in fossil fuel emissions and the large sensitivity of the carbon uptake estimates to those errors (Ballantyne et al., 2015), suggests that despite being thought to be the best-known term in the global carbon budget, systematic errors in fossil fuel emission estimates limit our assessment of the natural fluxes and the ocean-land partitioning from this inversion ensemble."*

**References:**

Ballantyne, A. P., R. Andres, and R. Houghton. 2015. "Audit of the Global Carbon Budget: Estimate Errors and Their Impact on Uptake Uncertainty." http://www.biogeosciences.net/12/2565/2015/bg-12-2565-2015.pdf .

Hansen, M. C., P. V. Potapov, R. Moore, M. Hancher, S. A. Turubanova, A. Tyukavina, D. Thau, et al. 2013. "High-Resolution Global Maps of 21st-Century Forest Cover Change." Science

342 (6160): 850–53.

Raymond, Peter A., Jens Hartmann, Ronny Lauerwald, Sebastian Sobek, Cory McDonald, Mark Hoover, David Butman, et al. 2013. "Global Carbon Dioxide Emissions from Inland Waters." Nature 503 (7476): 355–59.

Resplandy, L., Keeling, R. F., Rödenbeck, C., Stephens, B. B., Khatiwala, S., Rodgers, K. B., Long, M. C., Bopp, L., Tans, P. P. Revision of global carbon fluxes based on a reassessment of oceanic and riverine carbon, 11, 504—509, Nature Geoscience, 2018.

**Review 2:**

**General comments:** This paper uses the HIPPO dataset to evaluate recent atmospheric CO2 inversions and compares the spread of fluxes from the inversions with earlier inversion results and the Global Carbon Project budget. It is a useful presentation of the current state of inversions but I wonder whether more could have been done to increase our understanding of why the current inversions are more convergent than the earlier ones. I also wonder whether more information could have been extracted from the comparison with the HIPPO data. While these two points could be explored extensively, clearly some choices will have to be made as to what should be added to this paper and what should just be highlighted as areas for potential future work. I will try and provide some more specific suggestions below.

**Specific comments:**

**Use of HIPPO data:** the analysis presented in the paper effectively reduces the HIPPO data to 3 numbers, the 2009-2011 annual, JFM and JAS mean northern extratropical vertical gradient, for comparison with equivalently sampled model output. I suspect this allows model errors to cancel out in the averaging and makes it harder for the HIPPO data to discriminate between inversions. This is seen even in the seasonal results; the one inversion that matches the JAS gradient is also the one inversion that does not match the annual mean gradient. This raises the question around what value is put on matching the annual mean if seasonal means are not correct. This should at least be more fully discussed in the paper.

**Authors' response:** *We agree, and we have added several sentences highlighting this: "There is a significant relationship between HIPPO and the land only zonal average vertical gradient and both are correlated with the T + SET fluxes. This suggests that transport errors may be more critical in the summer season or that other factors compensate to obscure the relationship for these relatively coarse time averages in other seasons and for the annual means. While additional insights into model behavior could be gained from more detailed comparisons to individual models or in more controlled inversion ensembles, the varied nature of these inversion systems makes detailed analyses more challenging and beyond the scope of our current study."*

**Impact of harmonic fitting:** Fig S1 shows observations that are not well fitted by the harmonic, and this flows through into Fig 1 where the average HIPPO fit can be a long way from the mean observation point. Is this also the case for the model samples? If they were plotted on Fig 1 or Fig S1, would they scatter around the HIPPO observation point (e.g. the low point at day ~240) or would they scatter around the fitted lines? Is there information in the outliers to the fit which the models are able to capture?

**Authors' response:** *Please find below the Fig. 1 with the model value for each HIPPO slice. When the modelled fit lines do not match the individual slices, it is also the case for the observations. For example, the models capture that the summer drawdown (at least in 2011) is greater than can be fit by 2 harmonics. The results from the fits are not significantly different than from the points with ACTM (CDIAC) the largest vertical gradient in summer, followed by CAMS (v16r1) and ACTM (IEA); while ACTM (IEA) and ACTM (CDIAC) have the largest vertical gradients in winter.*

[Figure]

Figure: Same as Figure 1, but with model estimates of the 9 atmospheric transects.

It could be useful to create scatter plots of each binned HIPPO gradient against each equivalent model generated gradient. Do they cluster on a one-to-one line and with what correlation? Do different models give different scatter patterns/correlations? If these figures are informative, perhaps they could be added to the supplementary information.

**Authors' response:** *Please find below scatter plots of each binned HIPPO gradient against each equivalent model generated gradient. Qualitatively, there is no additional information.*
*The correlation between the modelled and observed vertical gradients for each slice showed excellent correlations, Pearson's coefficient ranging from r=0.94 to r=0.99.*

[Figure]

Figure: Model-specific scatter plots of the model (y axis) versus HIPPO observations (x axis) for the LT-UT vertical gradients, separately averaged for the 9 atmospheric transects in each panel.

My main concern is whether fitting the harmonic is minimising differences between the models and making it harder to discriminate between them? The low sensitivity to synoptic modelling bias presented in the Supplement would tend to confirm this. Would Fig 2a look different if instead of taking the annual mean from the harmonic fit, you just averaged the 9 data points - it wouldn't necessarily be representative of an annual mean but as long as the observations and models were treated in the same way that probably isn't important.

**Authors' response:** *To assess this we first calculated the annual mean vertical gradients by averaging the harmonic fits (as in the paper), or by averaging the 9 individual campaign vertical gradients (filtered with the same quality flags used for the fits) as suggested. The correlation across models between the annual means of the fits and the 9-slice averages is quite good, with $r^2$ = 0.75. We found that despite gradients being larger when averaging the 9 slices, the model spread is less with a standard deviation of 0.11 ppm for the averages of the fits, and 0.09 ppm for the slice averages, and a range of 0.37 for the averages of the fits, and 0.27 ppm for the slice averages. Therefore, our approach to average the fits is not minimizing the differences.*

**Convergence of current inversions compared to previous ones:** it would be good to provide some additional discussion in the paper as to why the current inversions are likely to be more convergent in their land estimates than the T3L2 ones. Since the T3L2 inversions all used the same method and priors, it would suggest that either there has been convergence in the transport models or that current methods are less sensitive to the atmospheric data and are more constrained by their priors. This seems possible if current methods are all using data at observed times rather than monthly means and not solving for large regions. This information is not currently in the paper. It would be good to add it (at least to supplementary material) i.e. the inversion method used, and some indication of the number of flask/in-situ atmospheric sites used and how they are used e.g. at measurement time, with what selection in the in-situ case. It would also be helpful to know what the magnitude of the prior land and ocean fluxes are for 2009-2011. This could either be provided as part of the model information in the supplement or perhaps plotted similarly to Fig 3a and Fig 4a. Knowledge of the prior may be particularly useful for understanding the land-ocean partition as I'm guessing it may contribute to some of the inversion differences.

**Authors' response (on the fluxes intercomparison):** *We added some information in the supplement, including the number of assimilated sites and if data was assimilated at discrete times or averaged. Almost all the systems are assimilating the data at discrete times (or hourly averaged) rather than monthly means. The 2 ACTM simulations are the only ones using monthly average data. Overall, there are many different methodological choices in each system and a thorough analysis of each possible contributor is beyond the scope of this study; we focused more on the information about actual fluxes that can be gleaned from the aggregate mean and spread. We suspect that the increasing density of assimilated observations, improved prior fields, and data assimilation methods all contribute to the convergence of the inversions, but also importantly transport error reduction and reanalyses datasets have significantly improved (see e.g. Bauer et al., 2015), so we added more discussion in Sect 3.1 on this topic:*
*"It is worth noting that reanalyses of meteorological observations have noticeably improved thanks to a better representation of unresolved processes in global models, improved data assimilation methods, and the increasing availability of satellite data, which makes the reanalyses perform better in the 2000s than for the 1990s and earlier (e.g., Gelaro et al., 2017; Bauer et al., 2015). As an example, the assimilation of new observations from the constellation of COSMIC global positioning system radio occultation (GPSRO) satellites has led to a significant improvement in meteorological analyses and forecasts (e.g., Healy, 2008)."*

**Fossil emissions and atmospheric growth rate:** I'm not sure I agree with some of the discussion around Fig. 4d. For example, the sentence on p15, line 17 starting 'One might expect ...' and 'counterintuively' (p18, line 5). The intent of an atmospheric inversion is to constrain fluxes by

the atmospheric data. Thus I would expect the AGR to be well fitted and any difference in fossil between inversions to be compensated by variations in the other fluxes. This is what you see in Fig 4c. Hence a near horizontal line (as in Fig 4d.) for the modelled WAGR is what I'd expect from an inversion perspective.

I don't think the suggestion of an opposite relationship (p15, line 19 'generally the opposite is true', also p18 line 5 'lower AGR') is strong enough to make this point, rather that the WAGR is insensitive to the FF estimate used.

If we were looking at different timeslices with real variation in fossil emissions (rather than just a variety of FF estimates for the same period) then it might be reasonable to expect a relationship between FF and WAGR but not when we are looking at a fixed period (2009-2011).

**Authors' response:** *Thank you for pointing this out. There is indeed no significant relationship between the total flux (WAGR) and FF and we agree a flat line is expected. We deleted the following sentence:*
*"One might expect that models with weaker fossil fuel emissions (to the right in Fig. 4D) would retrieve weaker atmospheric growth rates, but this figure shows that generally the opposite is true."*
*We also removed the following statement: "and the counterintuitive spread away from the mass balance line in 4d both"*

In general I found the 'WAGR', 'full AGR' and 'AGR' terminology confusing. Perhaps it is best to just use 'total flux' instead of WAGR since this is what you actually use from the inversion. Alternatively, it may be reasonably easy to calculate the MBL AGR from the inversions since I'd expect each inversion uses these sites and would likely keep information about their posterior fit to these sites. This measure could then be used as a direct comparison to the GCP calculated AGR and would be a useful additional indication of inversion spread.

**Authors' response (on the fluxes intercomparison):** *We agree that this calculation would be helpful, but because it would require reproducing the methodology of Dlugokencky and Tans for every model, we have not attempted it here. We did however note in the end of Sect 3.3 that in the most recent version of CarbonTracker, the agreement between MBL AGR and Total Flux is remarkably close suggesting that interannual variations in tropospheric and strat/trop mixing are not big contributors to interannual variability in total flux in other inversions. By our definition, and simple mass balance, WAGR = Total Flux. However, because the concept of AGR is so tightly linked to MBL AGR in people's minds we are happy to simply discuss Total Flux, and point out where others equate MBL AGR and Total Flux, and the inherent assumptions that entails.*
*In section 3.3 Variation in retrieved global carbon budgets, we replaced each occurrence of "WAGR" by "total flux", starting from its definition:*
*P13L7: From "For the models, the whole atmosphere growth rate (WAGR) is calculated as the subtraction of the ocean and land sink from the FF emissions." Becomes;*
*"For the models, the total flux is calculated as the subtraction of the ocean and land sink from the FF emissions. Note that by mass conservation the total flux equals the whole-atmosphere growth rate (WAGR), but that WAGR may differ from the MBL atmosphere growth rate (AGR) defined by surface stations, because of sampling biases or interannual variability in tropospheric mixing or stratosphere-troposphere exchange. GCP uses the MBL AGR as an estimate of Total Flux, and*

*assigns uncertainty of ± 0.19 PgCyr⁻¹ (Le Quéré et al.; 2016) for recent decades, with speculation that the relative uncertainty should decrease when averaging multiple years."*

Given my comments above, I think you need to be more careful with some of the messages that you draw out of the paper in the title, abstract and summary sections.
Specifically: (a) 'diverging on fossil fuel and atmospheric growth rate' as used in the title - diverging compared to what? The converging vs diverging language implies that the fossil and AGR terms now have larger uncertainty than the land term. Using the numbers in Table 2 and Table 3, it would be fair to say that choices about fossil prior now have as large an uncertainty as those retrieved for tropical land from the inversions, but both are still larger than that for WAGR. Perhaps this is the main point to be made - that reductions in the inversion spread for land and ocean fluxes, now means that more care needs to be taken with how the fossil term is included in the inversion. Likewise, in the abstract, p2, line 3, I don't agree that fossil and AGR terms 'dominate' the model spread since their uncertainties in Table 3 are smaller than those for land and ocean. Perhaps 'contribute to model spread at the largest scales and thus our ability to assess ...' Similarly at p18, line 18.

**Authors' response:** *We agree that the "diverging" term was confusing. By "converging" we meant that over time the inversions are getting closer together, but by "diverging" we really meant "disagreeing with each other" (with no implied temporal trend). We have changed "diverging" to "disagreeing" in the title. We also agree that the land and ocean terms still have the largest uncertainties, but these are retrieved quantities whereas FF is prescribed and AGR is closely linked to the optimized-against concentrations, so our expectations of agreement on FF and AGR are higher than for ocean and land. We have removed the "dominate the model spread at the largest scales and" clause in 3 places.*

(b) The abstract makes a general statement about agreement with HIPPO (p1, line 9) but the paper only really presents the northern vertical gradient so perhaps this statement should be modified to only include the 'in particular ...' phrase (unless any of the analysis suggested above provides further insight into the agreement with HIPPO data across inversions)

**Authors' response:** *The sentence has been updated to: "Most modelled CO₂ fields agree reasonably well with the HIPPO observations, specifically for the annual mean vertical gradients in the northern hemisphere."*

(c) Abstract, p1, line 15. I am not convinced that the large disagreements on ocean-land partitioning are strongly influenced by the prescribed fossil fuel and associated differences in retrieved AGR.

**Authors' response:** *We updated the sentence to remove the second clause and it now reads: "However, models still have large disagreements on ocean-land partitioning."*

**Technical comments**
p1, line 6: 'latitudinal distribution' - is this the best term to use since the paper really only focusses on the split between north and tropics+south.
**Authors' response:** *We change it to "broad latitudinal separation"*

p2, line 19: I found the 'NE' and 'SE' acronyms distracting as I had to stop myself from reading them as North-East and South-East. Perhaps just use 'N/S' or 'NEx/SEx'
**Authors' response:** *We have changed to use NET/SET instead throughout the manuscript.*

p3, line 34: The GCP2016 acronym is used here but not defined until p4.
**Authors' response:** *This has been corrected, the acronym is defined on p3 for its first use.*

p4, line 9: missing ')' after Boden et al 2016.
**Authors' response:** *This has been corrected.*

p4, line 10: You note the possibility of using the ACTM inversions to assess the sensitivity to the FF, but it wasn't obvious to me that you actually do that within the analysis presented in the paper.

**Authors' response:** *We added the following sentences in Sect. 3.1:*
*"The ACTM-IEA inversion is an outlier and overestimates by 1 ppm the winter season average vertical gradient. For ACTM, the global annual IEA emissions are less than CDIAC (Fig. 4C and D) which results in a weaker northern extratropical sink (Fig. 2A and 3A) which corresponds with a more positive LT-UT northern extratropical gradient (Fig. 2A and S2) and a more positive N-S gradient (Figure S2), comparing just the 2 ACTM versions. Differences across inversion systems in Figure S2 also depend also on transport and inversion scheme and the resulting spatial distribution of sources and sinks."*

p6, line 20: It would be good to note somewhere here that the model is sampled along flight tracks i.e. move the comment from p7, line 2-3 earlier.

**Authors' response:** *We updated the sentence earlier in the same section: "To do this, we first detrend the observations and model sampled along the flight-track output by subtracting a deseasonalized and smoothed long-term trend record from Mauna Loa,…"*

p7, line 4: suggest replace 'curtain averages' with '150 W transect'
**Authors' response:** *Done*

p7, line 5: Add '(Section 4)' after 'supplement'
**Authors' response:** *We have edited this to read "in Section 4 of the supplement."*

p8, line 8: perhaps add 'and model' before 'vertical gradients'
**Authors' response:** *Done*

p8, line 9: Should this be -2.24?
**Authors' response:** *Yes, it has been corrected, thanks.*

p8, line 11: It's not clear to me what number is being quoted here. A range of 3.54 seems to match the Fig 2b, but what does the 1.04 refer to? It seems unusual to quote an uncertainty on a range. Perhaps just give the mean and 1-sigma as this would then be comparable with the number given in line 9.

**Authors' response:** *We corrected this to reflect the mean and spread of the land sink. The (3) whole sentences now read:*
*The mean and relative spread of 10 simulations for the posterior annual mean NE land flux is -2.24 PgC/yr ± 0.29 PgC/yr (13 %, 1σ). Aside from the ACTM-IEA simulation, all models are within the uncertainty range of 0.15 ppm or 50 % of the measured vertical gradient. This contrasts to the TransCom3 Level 2 simulations which had an annual mean of -2.42 PgC/yr ± 1.05 (43 %) PgCyr⁻¹ for NE land flux and disagreed with the observed vertical gradient by ~0.5 ppm on average and as much as 1.3 ppm (186 %).*

p8, line 12: 1.3 ppm - this is from one outlier, perhaps better to give the typical mismatch ~0.5 ppm.
**Authors' response:** *Done, see previous response.*

p9. line 12: Perhaps 'Results shown in Fig. 2d and Fig. S7 are consistent.'
**Authors' response:** *This sentence has now been deleted (see above).*

p9, line 24: Insert 'of' before 'retrieved land fluxes'
**Authors' response:** *Done*

p9, line 25: what did you mean by 'and on 2.'?
**Authors' response:** *The end of the sentence has been changed to: "as shown in Fig. 3 and Fig. 2."*

p9, line 27: Add 'from the GCP2016 estimate' after 'disagreement on the total land sink'
**Authors' response:** *Done*

Figure 3 caption: Is this complete? There was no specific information about panel D. Table 2: The T+SE land flux is the same (0.34 +/- 0.27) for RECCAP Group 1 and This Study. Is this correct?
**Authors' response:** *It is correct, both ends up having the same value (but not for the Northern Extratropical flux). We added "Panel D shows the results from our new set of models for the period 2004-2014".*

p13, line 13: 'full AGR'. If this is intended to be the same as the WAGR, then just use WAGR.
**Authors' response:** *Full AGR and WAGR have been replace by total flux in the manuscript.*

p14, Figure 4 caption. line 3 'We' not 'we'. line 6 'Here' not 'He'.
**Authors' response:** *Done*
I was confused by the use of 'WAGR-FF line' and 'WAGR line' in line 8. If these are for GCP, are they from AGR not WAGR?
**Authors' response:** *This is now defined as the total flux.*

p15, line 3: 'Because of the intentionally different FF source'. I don't think this is the explanation for the difference between ACTM-CDIAC and TM5-4DVar, since their FF values seem relatively similar (~9 PgC/y). Perhaps the different prior ocean flux used makes a contribution.

**Authors' response:** *We updated the sentence to: "Taking the 2 extreme models the ACTM-CDIAC and TM5-4DVar estimates provide very different distributions of fluxes. ACTM-CDIAC suggests stronger land sinks, both over the NET and the T + SET regions, and a lower ocean sink while TM5-4DVar suggests the opposite. This leads to a range of around 2 PgC/yr on the model ocean sink. Because of an intentionally different FF source, but with the same inversion system, the ACTM-CDIAC and ACTM-IEA retrieved land fluxes differ by slightly less than 1 PgC/yr and ocean fluxes that differ by half a PgC/yr."*

p15, line 16: 'again defined' - perhaps qualify this as just for the models, assuming that the GCP line is from the MBL AGR.
**Authors' response:** *This is now defined as the total flux.*

p15, line 25-26: 'counterintuitive spread away from the mass balance line in Fig 4d' - does this require the assumption of a constant airborne fraction? If so, it might be good to note this.
**Authors' response:** *We removed this sentence.*

Figure 5 is difficult to read. Perhaps plot across a smaller range, or plot as differences from GCP. I assume that panel B is basically the same information as the y axis of Fig 4d just as a sum over 3 years rather than the mean. Panel B is not described in the Figure caption. Replace 'full AGR' with 'WAGR' in figure caption.

**Authors' response:** *The year to year variability in GCP is also of interest so we have not plotted as differences. We replaced 'full AGR' with 'total flux' and have added the following description: "Panel B shows the sum of the total flux for the three years (2009 to 2011)".*

CAMS looks slightly smaller than GCP in this figure but slightly larger in Fig 4d.

**Authors' response:** *Thanks for noticing it, GCP is lower than CAMS, and this was an issue on the panel B of Fig. 5. which we have now corrected.*

p18, line 5: 'lower' instead of 'slower' - though I'm not sure the signal is strong enough to really make this point - ellipse on Fig 4d is close to horizontal.
**Authors' response:** *Corrected to lower instead of slower.*

p18, line 7: not sure where the +/- 0.05 number comes from and exactly what it means.
**Authors' response:** *Thanks for catching this mistake. It should have read 0.10 ppm, and comes from the analysis of CT2017 results in the last paragraph of Section 3.3.*

**Supplementary material**

p1: define STL
**Authors' response:** *The sentence now reads: "We subtract a deseasonalized smoothed trend component from a seasonal trend decomposition using Loess (STL, Cleveland et al. 1990) fit,"*

p1: how different are the MLO reference trends subtracted from the models? I would assume they are quite similar since the inversions would do a good job at fitting the MLO data, but it might be worth mentioning.

**Authors' response:** *We choose to subtract the same trend from all models, so that the comparison is not sensitive to how MLO is modelled. The end of the sentence is: "fit of the Mauna Loa Observatory in-situ measurement time series to provide a common reference for both observations and models."*

p1: You might like to add a comment interpreting the results presented in Fig S2 and S3.

**Authors' response:** We added: *"These figures show that the latitudinal distribution is well represented in the models while more differences are seen in the vertical."*

p2: The text says your averaging was for 5 degree bins but the example given seems to be for a 10 degree bin. Since most of your analysis only uses 20-87N and below 400 hPa perhaps it would be simpler to only plot these regions in panel 1 and 3.

**Authors' response:** *We now just plot the data between 20 to 87N on panel 3.*

p3 and p4: In Fig S2 and S3 it would be helpful to put all the CT cases in one row, and put the ACTM cases next to each other. I thought the ACTM cases looked surprisingly different in Fig S2.

**Authors' response:** *Done, the ACTM cases look different because there is an offset of $CO_2$ (that shifts the colors), the spatial structure is actually not that different. The ACTM-CDIAC model has a weaker annual mean vertical gradient as discussed in more detail in our response to Reviewer 1.*

Inversion descriptions: Jena, biosphere and fires: 'Constant' - do these priors include a mean seasonal cycle?

**Authors' response:** *The biosphere and fires component are set to zero (see Rödenbeck et al. 2018).*

**References:**

Bauer, P., Thorpe, A., and Brunet, G.: The quiet revolution of numerical weather prediction, Nature, 525, 47–55, https://doi.org/10.1038/nature14956, 2015.

R. B. Cleveland, W. S. Cleveland, J.E. McRae, and I. Terpenning (1990) STL: A Seasonal-Trend Decomposition Procedure Based on Loess. Journal of Official Statistics, 6, 3–73.

Gelaro, R., McCarty, W., Suárez, M. J., Todling, R., Molod, A., Takacs, L., Randles, C. A., Darmenov, A., Bosilovich, M. G., Reichle, R., Wargan, K., Coy, L., Cullather, R., Draper, C., Akella, S., Buchard, V., Conaty, A., da Silva, A. M., Gu, W., Kim, G.-K., Koster, R., Lucchesi, R., Merkova, D., Nielsen, J. E., Partyka, G., Pawson, S., Putman,W., Rienecker, M., Schubert, S. D., Sienkiewicz, M., and Zhao, B.: The Modern-Era Retrospective Analysis for Research and Applications, Version 2 (MERRA-2), Journal of Climate, 30, 5419–5454, https://doi.org/10.1175/JCLI-D-16-0758.1, 2017.

Healy, S. B.: Forecast impact experiment with a constellation of GPS radio occultation receivers, Atmospheric Science Letters, 9, 111–118, https://doi.org/10.1002/asl.169, 2008.

Rödenbeck, C., Zaehle, S., Keeling, R., and Heimann, M.: How does the terrestrial carbon exchange respond to inter-annual climatic variations? A quantification based on atmospheric $CO_2$ data, Biogeosciences, 15, 2481-2498, https://doi.org/10.5194/bg-15-2481-2018, 2018.

Saeki, T. and Patra, P. K.: Implications of overestimated anthropogenic CO2 emissions on East Asian and global land CO2 flux inversion, Geoscience Letters, 4, https://doi.org/10.1186/s40562-017-0074-7, 2017.